# Simplified Estimation Method for Maximum Deflection in Bending-Failure-Type Reinforced Concrete Beams Subjected to Collision Action and Its Application Range

**Yusuke Kurihashi * and Hiroshi Masuya**

Department of Geosciences and Civil Engineering, Kanazawa University, Kakuma-machi,
Kanazawa 920-1192, Japan; masuya@se.kanazawa-u.ac.jp
* Correspondence: kuri@se.kanazawa-u.ac.jp; Tel.: +81-76-234-4602

**Abstract:** As natural disasters have become increasingly severe, many structures designed to prevent rockfalls and landslides have been constructed in various areas. The impact resistance capacity of a reinforced concrete (RC) rock shed can be evaluated using its roof deflection. This study establishes a method for estimating the maximum deflection of a bending-failure-type RC beam, subjected to collisions that is based on the energy conservation concept—in which, the transmitted energy from a collision is equivalent to the energy absorbed by the beam. However, the following assumptions have never been confirmed: (1) The energy transmitted to the RC beam, due to the dropped weight, can be estimated by assuming a perfect plastic collision; and (2) the energy absorbed by the RC beam can be estimated by assuming plane conservation. In this study, these assumptions were verified using 134 previous test results of RC beams subject to weight collisions. In addition, we proposed a simple method for calculating the maximum deflection and its application scope. With this method, a performance-based impact-resistant design procedure for various RC structures can be established in the future. Moreover, this method will significantly improve the maintenance and management of existing RC structures subject to collisions.

**Keywords:** RC beam; impact loading test; maximum deflection; energy conservation concept

## 1. Introduction

In recent years, natural disasters have become more severe because of climate change caused by global warming. Torrential rains have occurred in various parts of the world, causing large-scale slope disasters [1–3]. Many rock sheds, retaining walls, and barriers are used in coastal and mountainous areas as road disaster prevention countermeasures. Figure 1 shows examples of rockfall and landslide disasters in rock sheds in coastal and mountainous regions. Instead of being designed with specification-based approaches, such as the allowable stress method, these protective structures should be designed using performance-based design methods [4,5]. In particular, the impact resistance capacity of a reinforced concrete (RC) rock shed can be evaluated using the deflection of its roof [6–8], which can be used to set each limit state of the shed. However, even for basic structural members, such as an RC beam, an appropriate method for estimating the maximum deflection has not yet been established. Hence, many research institutes were attempting to establish a method for estimating the maximum deflection of RC beams subjected to collision action [9–14]. Figure 2 illustrates the general impact loading method and an example of the test results.

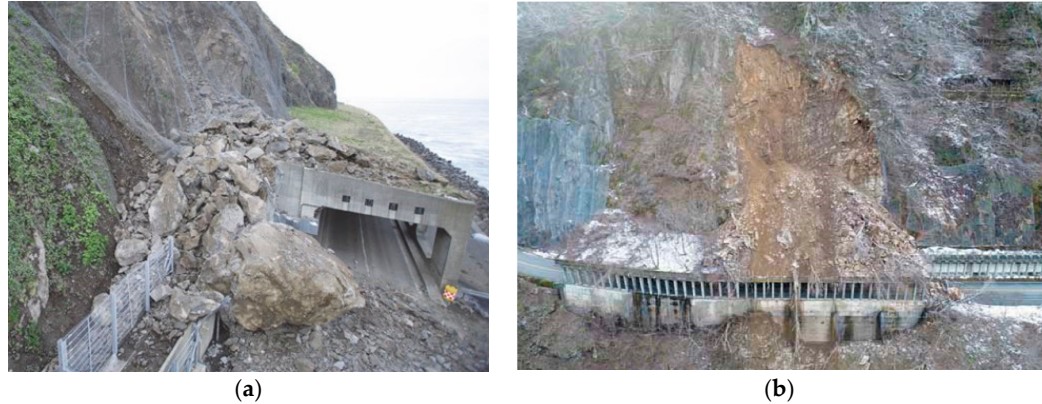

**Figure 1.** Examples of rockfall and landslide disaster; (**a**) Rockfall in a coastal area (Hokkaido, Japan, 2008), (**b**) Landslide in a mountainous area (Ishikawa, Japan, 2018).

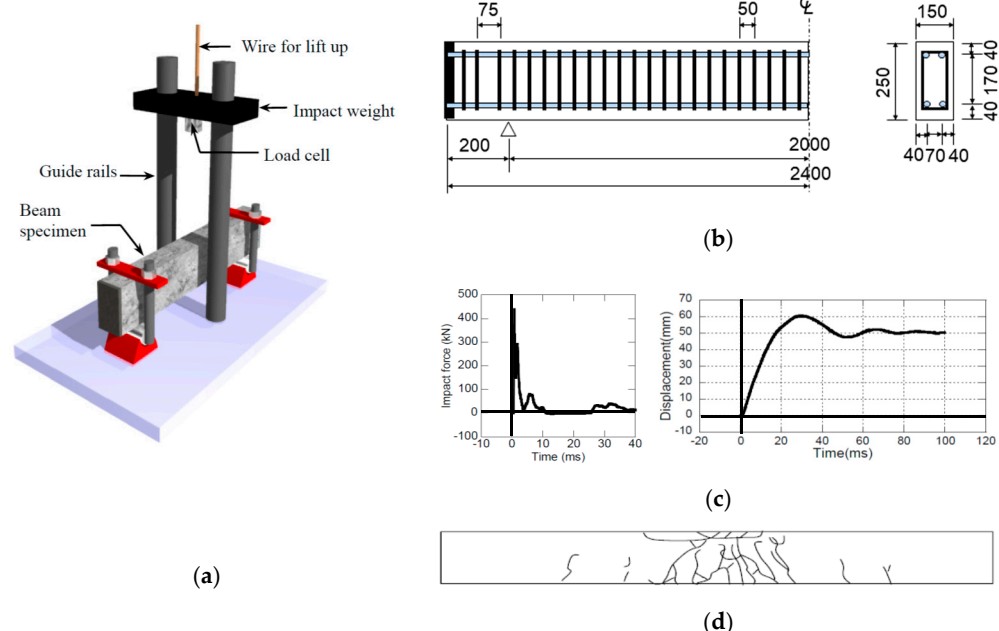

**Figure 2.** An example of a weight-falling impact test and its results; (**a**) Setup, (**b**) dimensions of the RC beam, (**c**) time histories of impact force and deflection, (**d**) crack pattern after the experiment.

In previous studies, Kishi et al. proposed a method for estimating the maximum deflection of RC beams based on the results of their weight-falling impact tests under bending failure [9]. This estimation method was suggested based on the linear relationship between the kinetic energy of the weight and the maximum deflection of the RC beam. This method was also summarized in Structural Engineering Series 22, published by the Japan Society of Civil Engineers (JSCE) [10]. Tachibana et al. conducted impact loading tests of RC beams and proposed an estimation equation for the maximum deflection [11]. Fujikake et al. calculated the load–deflection relationship of RC beams, considering the strain rate effect of concrete and reinforcing bars and subsequently attempted to estimate the maximum deflection based on the conservation laws of momentum and energy. Their study clarified that the experimental results presented by other researchers can be evaluated conservatively [12]. Kishi et al. proposed a residual deflection estimation method for large RC beams with a clear span of 8.0 m [13]. In this estimation method, a correction formula based on the mass ratio of the weight and RC beam was empirically introduced.

Recently, Hwang et al. proposed a maximum deflection estimation method based on the conservation law of energy, considering (i) input energy (due to falling weight), (ii) energy loss at the

time of collision, (iii) change in potential energy (due to the deflection of the RC beam), (iv) energy absorption (due to the bending deformation of the beam), and (v) energy loss (due to the peeling of concrete at the upper edge of the beam) [14]. Their results showed that the maximum deflection of RC beams can be estimated based on previous experimental studies. Furthermore, a numerical analysis using the finite element method was conducted [15–17].

However, few studies have focused on the simple estimation method for maximum deflection, and its application range for cases where a bending-failure-type RC beam is plastically deformed to absorb energy. The simple calculation method enables the appropriate maintenance based on the reliability design of existing disaster prevention structures, taking into consideration the uncertainty of action and variations in material strength [18,19].

Currently, the energy transmitted to the RC beam during a weight collision can often be calculated under the assumption of a perfect plastic collision, in which the weight and RC beam move together without repulsion after the collision [12,14]. However, no investigation exists where the validity of this assumption is verified by the impact test results of RC beams under different conditions. In addition, the energy absorbed by the RC beam is calculated to be the area surrounded by the curve by obtaining the RC beam load–deflection relationship under the assumption of plane conservation [9–14]. However, the applicable range of this assumption has not been clarified.

In this study, these assumptions were verified based on the experimental results of 134 cases conducted in previous studies, and a simple calculation method for the maximum deflection and its application range were proposed. Based on the proposed method for estimating the maximum deflection of a bending-failure-type RC beam, an impact-resistant-design procedure for various RC structures can be established in the future. Moreover, the proposed method will significantly improve the maintenance and management of existing RC structures subject to a collision action.

## 2. Outline of the Simple Estimation Method of Maximum Deflection for Bending-Failure-Type RC Beams Subject to a Weight Collision

### 2.1. Energy Conservation Concept

As previously discussed, the maximum deflection is calculated as the deflection when the transmitted impact energy $E_t$ and the absorbed energy $E_a$ of the RC beam are equivalent, based on the law of conservation of energy.

$$E_t = E_a \tag{1}$$

In addition to the above energy, it is possible that the energy generated by the movement of the RC beams and the energy generated by the scattering of concrete pieces may have an effect. However, these effects are extremely small compared to $E_t$ and $E_a$; therefore, they were excluded from this study. The calculation methods for $E_t$ and $E_a$ are detailed below.

### 2.2. Calculation Method of Transmitted Impact Energy $E_t$

The energy $E_t$ transmitted to the RC beam during a weight collision was calculated under the assumption of a perfect plastic collision where the weight and the RC beam moved together without repulsion after the collision. The formula for calculating $E_t$ was derived, as follows.

First, the momentum conservation laws, immediately before and after the weight collides with the beam, are expressed as follows:

$$M_w V = (M_{be} + M_w) V_a \tag{2}$$

$$V_a = (M_w/(M_{be} + M_w)) V \tag{3}$$

where $M_{be}$ is the equivalent mass of the beam obtained by assuming that the vibration mode of the beam is equivalent to the first-order bending mode and multiplying the mass of the beam within its

clear span $M_b$ by 17/35. Furthermore, $M_w$ is the mass of the impact weight, $V$ is the velocity of the weight immediately before collision, and $V_a$ is the velocity of the mass point, including the falling weight and the beam, immediately after the collision. Subsequently, we considered the kinetic energy before collision $E_k$ can be estimated using Equation (4), and the energy after collision $E_{ka}$ can be derived using Equations (5)–(7) as follows:

$$E_k = 1/2 \; M_w \; V^2 \text{ (before collision)} \tag{4}$$

$$E_{ka} = 1/2 \; (M_{be} + M_w) \; V_a{}^2 \text{ (after collision)} \tag{5}$$

$$= M_w{}^2 / (2 \; (M_{be} + M_w)) \; V^2 \tag{6}$$

$$= (M_w/(M_{be} + M_w)) \; E_k \tag{7}$$

where $E_{ka}$ is the kinetic energy of the combined weight and beam immediately after the collision and corresponds to the energy transmitted to the beam $E_t$. The energy $E_t$ can be determined using Equation (8).

$$E_t = (M_w/(M_{be} + M_w)) \; E_k \tag{8}$$

$$M_{be} = (17/35) \; \rho A L \tag{9}$$

where $\rho$ is the unit mass of the RC beam (=2.5), $A$ is the sectional area of the beam, and $L$ is the clear span of the beam.

### 2.3. Calculation Method for the Absorbed Energy $E_a$ of the RC Beam

$E_a$ is calculated as the area under the load–deflection curve obtained by the fiber model, considering the plane conservation of the beam section.

$$E_a = \int_0^\delta P(\delta)d\delta \tag{10}$$

The calculation procedure is outlined as follows: (i) Divide the height of the section into 5 mm intervals and along the span direction into 100 mm intervals, considering the solution stability; (ii) increase the upper-edge strain by 10 μ to determine the cross-sectional neutral axis at each stage, and determine the curvature-bending moment relationship; (iii) determine the bending moment distribution along the span direction at each load step and the corresponding curvature distribution; and (iv) calculate the deflection of the span center using the elastic load method. Figure 3 shows the concept of the fiber model.

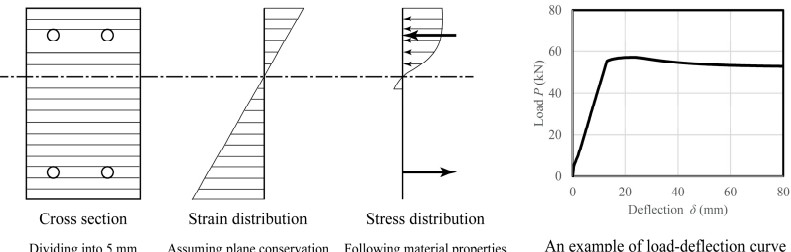

Cross section — Strain distribution — Stress distribution — An example of load-deflection curve
Dividing into 5 mm — Assuming plane conservation — Following material properties

**Figure 3.** Calculation concept of the fiber model.

The constitutive material laws for concrete and reinforcing bars were determined, as shown in Figure 4, following the JSCE Concrete Standards Design [20].

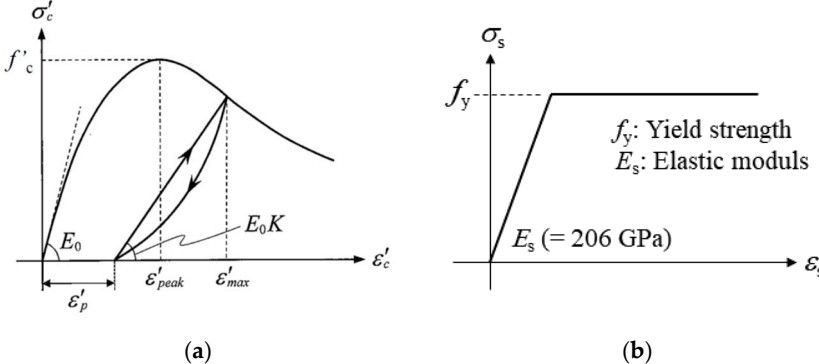

**Figure 4.** Stress-strain relationships of concrete and rebar for fiber model calculation, (**a**) Concrete, (**b**) Steel rebar.

The compressive stress and strain relationships of concrete were defined as follows:

$$\rho E_c = E_0 K \left( \varepsilon\prime_c - \varepsilon\prime_p \right) \tag{11}$$

$$E_0 = \frac{2 \cdot f\prime_c}{\varepsilon\prime_{peak}} \tag{12}$$

$$K = \exp\left\{ -0.73 \frac{\varepsilon\prime_{max}}{\varepsilon\prime_{peak}} \left[ 1 - \exp\left( -1.25 \frac{\varepsilon\prime_{max}}{\varepsilon\prime_{peak}} \right) \right] \right\} \tag{13}$$

$$\varepsilon\prime_p = \varepsilon\prime_{max} - 2.86 \cdot \varepsilon\prime_{peak} \left[ 1 - \exp\left( -0.35 \frac{\varepsilon\prime_{max}}{\varepsilon\prime_{peak}} \right) \right] \tag{14}$$

Material test results were used to establish the compressive strength of concrete and yield strength of the reinforcing steel. The tensile strength $f_t$ of concrete was estimated using the following equation, according to JSCE [20]:

$$f_t = 0.23 \, f\prime_c{}^{\frac{2}{3}} \tag{15}$$

As the tensile fracture energy of concrete is substantially smaller than the absorbed energy owing to the bending plastic deformation of the RC beams, the tension softening of concrete was not included in the scope of this study. Moreover, perfect bonding was assumed between the rebar and concrete.

In this calculation, the strain rate cannot be considered because it changes significantly in time and space after a weight collision. It is difficult to apply it to a simple estimation formula. In addition, the applicable range of absorbed energy calculated under the assumption of plane conservation has not been clarified yet. This should be confirmed through a comparison with the numerous existing experimental results.

## 3. Verification by Comparison with Previous Experimental Results

### 3.1. Validity of a Perfect Plastic Collision

#### 3.1.1. Experimental Results Used for This Investigation

The validity of the assumption of a perfect plastic collision was evaluated using experimental results from previous studies [9,10]. Table 1 lists the specifications of the tested RC beams and experimental conditions. The clear span of the beam, *L*, is the distance between the two supports, and $p_t$ is the tensile rebar ratio. The numerical values for the calculated bending capacity $P_u$ and shear capacity $Q_u$ were obtained from the literature [9].

**Table 1.** List of test specimens used for analysis [9].

| Name | Size of Section | Rebar $\varphi$ (mm) | N | L (m) | $p_t$ (%) | $f'_c$ (MPa) | $f_y$ (MPa) | $P_u$ (kN) | $Q_u$ (kN) | $\alpha$ | $M_b$ (t) | $M_{be}$ (t) | V (m/s) | $M_w$ (t) | $E_k$ (kJ) | $E_t$ (kJ) |
|---|---|---|---|---|---|---|---|---|---|---|---|---|---|---|---|---|
| G1-1 | 200 × 300 | 19 | 2 | 3.0 | 1.10 | 33.7 | 379 | 69.6 | 195.4 | 2.81 | 0.450 | 0.219 | 7.00 | 0.3 | 7.35 | 4.25 |
| G1-1S | | | | | | | | | | | | | 7.00 | | 7.35 | 4.25 |
| G2-1 | | | | | | | | | | | | | 4.00 | | 2.40 | 1.84 |
| G2-2 | 150 × 250 | 13 | 2 | 2.0 | 0.80 | 32.2 | 373 | 38.1 | 139.7 | 3.67 | 0.188 | 0.091 | 5.00 | 0.3 | 3.75 | 2.88 |
| G2-3 | | | | | | | | | | | | | 6.00 | | 5.40 | 4.14 |
| G2L-1 | | | | | | | | | | | | | 4.00 | | 3.20 | 2.61 |
| G2L-2 | 150 × 250 | 13 | 2 | 2.0 | 0.80 | 32.2 | 373 | 38.1 | 139.7 | 3.67 | 0.188 | 0.091 | 5.00 | 0.4 | 5.00 | 4.07 |
| G2L-3 | | | | | | | | | | | | | 6.00 | | 7.20 | 5.86 |
| G3-1 | | | | | | | | | | | | | 4.00 | | 2.40 | 1.84 |
| G3-2 | 150 × 250 | 13 | 2 | 2.0 | 0.80 | 34.6 | 393 | 40.2 | 141.1 | 3.51 | 0.188 | 0.091 | 5.00 | 0.3 | 3.75 | 2.88 |
| G3-3 | | | | | | | | | | | | | 6.00 | | 5.40 | 4.14 |
| G4-1 | | | | | | | | | | | | | 4.00 | | 2.40 | 1.84 |
| G4-2 | 150 × 250 | 13 | 2 | 2.0 | 0.80 | 32.3 | 373 | 38.1 | 139.8 | 3.52 | 0.188 | 0.091 | 5.00 | 0.3 | 3.75 | 2.88 |
| G5-1 | | | | | | | | | | | | | 6.00 | | 7.20 | 4.66 |
| G5-2 | 200 × 300 | 19 | 2 | 3.0 | 1.10 | 39.2 | 379 | 70.4 | 200.4 | 2.85 | 0.450 | 0.219 | 7.00 | 0.4 | 9.80 | 6.34 |
| G6-1 | 250 × 250 | 19 | 2 | 2.0 | 1.09 | 34.7 | 392 | 87.4 | 191.4 | 2.19 | 0.313 | 0.152 | 5.00 | 0.3 | 3.75 | 2.49 |
| G7-1 | | | | | | | | | | | | | 5.00 | | 3.75 | 2.13 |
| G7-2 | 250 × 250 | 19 | 2 | 3.0 | 1.09 | 34.7 | 392 | 58.3 | 162.3 | 2.78 | 0.469 | 0.228 | 6.00 | 0.3 | 5.40 | 3.07 |
| G8-1 | 200 × 200 | 25 | 2 | 2.0 | 3.17 | 34.7 | 383 | 102.3 | 158.4 | 1.55 | 0.200 | 0.097 | 6.00 | 0.3 | 5.40 | 4.08 |
| G9-1 | | | | | | | | | | | | | 5.00 | | 3.75 | 2.83 |
| G9-2 | 200 × 200 | 25 | 2 | 3.0 | 3.17 | 34.7 | 383 | 68.2 | 136.3 | 2.00 | 0.200 | 0.097 | 6.00 | 0.3 | 5.40 | 4.08 |

Table 1. *Cont.*

| Name | Size of Section | Rebar $\varphi$ (mm) | N | L (m) | $p_t$ (%) | $f'_c$ (MPa) | $f_y$ (MPa) | $P_u$ (kN) | $Q_u$ (kN) | $\alpha$ | $M_b$ (t) | $M_{be}$ (t) | V (m/s) | $M_w$ (t) | $E_k$ (kJ) | $E_t$ (kJ) |
|---|---|---|---|---|---|---|---|---|---|---|---|---|---|---|---|---|
| G10-1 | | | | | | | | | | | | | 4.00 | | 2.40 | 1.49 |
| G10-2 | 200 × 250 | 19 | 2 | 3.0 | 1.36 | 23.5 | 404 | 56.6 | 289.3 | 5.11 | 0.375 | 0.182 | 5.00 | 0.3 | 3.75 | 2.33 |
| G10-3 | | | | | | | | | | | | | 6.00 | | 5.40 | 3.36 |
| G10-4 | | | | | | | | | | | | | 7.00 | | 7.35 | 4.57 |
| G11-1 | | | | | | | | | | | | | 3.13 | | 2.45 | 1.76 |
| G11-2 | | | | | | | | | | | | | 4.20 | | 4.41 | 3.16 |
| G11-3 | 200 × 300 | 22 | 2 | 2.7 | 1.55 | 23.6 | 401 | 94.4 | 164.8 | 1.66 | 0.405 | 0.197 | 5.05 | 0.5 | 6.38 | 4.58 |
| G11-4 | | | | | | | | | | | | | 5.78 | | 8.35 | 5.99 |
| G11-5 | | | | | | | | | | | | | 6.42 | | 10.30 | 7.39 |
| G11-6 | | | | | | | | | | | | | 7.00 | | 12.25 | 8.79 |
| G12-1 | 200 × 300 | 19 | 3 | 2.7 | 2.72 | 23.6 | 407 | 103.9 | 168.1 | 1.52 | 0.405 | 0.197 | 7.67 | 0.5 | 14.71 | 10.55 |
| G13-1 | 200 × 400 | 25 | 2 | 2.7 | 1.45 | 23.6 | 406 | 178.3 | 400.2 | 2.11 | 0.540 | 0.262 | 7.67 | 0.5 | 14.71 | 9.65 |
| G14-1 | 200 × 350 | 25 | 2 | 2.7 | 1.69 | 23.6 | 406 | 149.8 | 312.2 | 1.96 | 0.473 | 0.230 | 7.67 | 0.5 | 14.71 | 10.08 |
| G15-1 | 200 × 400 | 29 | 2 | 2.7 | 1.84 | 23.6 | 406 | 224 | 850.1 | 3.58 | 0.540 | 0.262 | 7.67 | 0.5 | 14.71 | 9.65 |
| G16-1 | 200 × 370 | 25 | 2 | 2.7 | 1.58 | 23.6 | 406 | 161.2 | 371.7 | 2.16 | 0.500 | 0.243 | 7.67 | 0.5 | 14.71 | 9.90 |

$\varphi$: Nominal diameter of the tensile rebar, $N$: Number of rebars, $L$: Clear span length of the beam, $p_t$: Tensile rebar ratio, $f'_c$: Compressive strength of concrete, $f_y$: Yield strength of the tensile rebar, $P_u$: Calculated bending capacity, $Q_u$: Calculated shear capacity, $\alpha$: Shear-bending capacity ratio (= $Q_u/P_u$), $M_b$: Mass of beam, $M_{be}$: Equivalent mass of beam, $M_w$: Mass of weight $V$: Impact velocity of weight, $E_k$: Kinetic energy of weight (input energy), $E_t$: Transmitted energy.

In all 36 experimental cases employed in this study, the steel weight was dropped only once from a predetermined height to the RC beam center. The RC beam was placed on a fulcrum with a lifting prevention jig. The boundary condition of the fulcrum was similar to that of the pinned support. The beams were all rectangular RC beams.

The cross-sectional width, height, and span length of the test specimens varied from 150 to 250 mm, 200 to 400 mm, and 2 to 3 m, respectively. Further, the tensile rebar ratio and mass of the falling steel weight varied from 0.8% to 3.17% and from 300 to 500 kg, respectively. Furthermore, the velocity of the falling weight before collision varied from 4 to 7.67 m/s.

### 3.1.2. Comparison Between the Estimated and Measured Maximum Deflections

The maximum deflection $\delta_u$, assuming a perfect plastic collision, was calculated as the satisfying deflection Equation (16) based on the energy conservation concept. For comparison, the maximum deflection $\delta_{uk}$, assuming no energy loss, was also calculated using Equation (17). Figure 5 shows a conceptual diagram of $\delta_u$ and $\delta_{uk}$ obtained with $E_t$ and $E_k$, respectively, using the calculated $P$–$\delta$ curve.

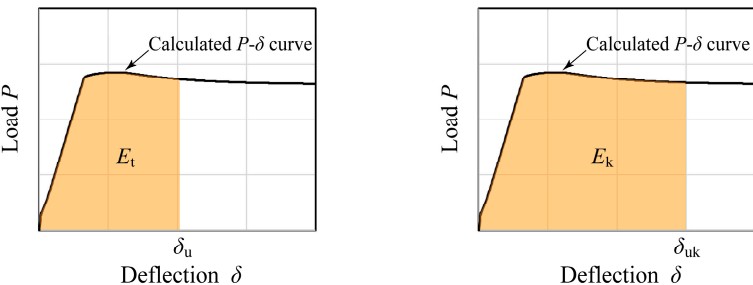

**Figure 5.** Conceptual diagram for estimating $\delta_u$ and $\delta_{uk}$.

From Figure 5, it can be seen that $\delta_u$ is smaller than $\delta_{uk}$. This means that in the case of considering a perfect plastic collision, energy is lost when a weight collides.

$$E_t \ = \ (M_w/(M_{be} \ + \ M_w)) \ E_k \ = \ \int_0^{\delta_u} P(\delta)d\delta \tag{16}$$

$$E_k \ = \ \int_0^{\delta_{uk}} P(\delta)d\delta \tag{17}$$

Figure 6 compares the estimated maximum deflections $\delta_{uk}$ and $\delta_u$ based on $E_k$ and $E_t$, respectively, with the experimental results of $\delta_{u.exp}$. The figure shows that the estimated maximum deflection $\delta_{uk}$ based on $E_k$ significantly exceeds the experimental value and that this difference increases when the deflection is larger. In contrast, the estimated maximum deflection $\delta_u$ based on the transmitted energy $E_t$ is generally larger than the experimental value; however, it is closer to the experimental value than in the case of $\delta_{uk}$. This indicates that the maximum deflection $\delta_{u.exp}$ can be determined accurately and conservatively by using the transmitted energy $E_t$.

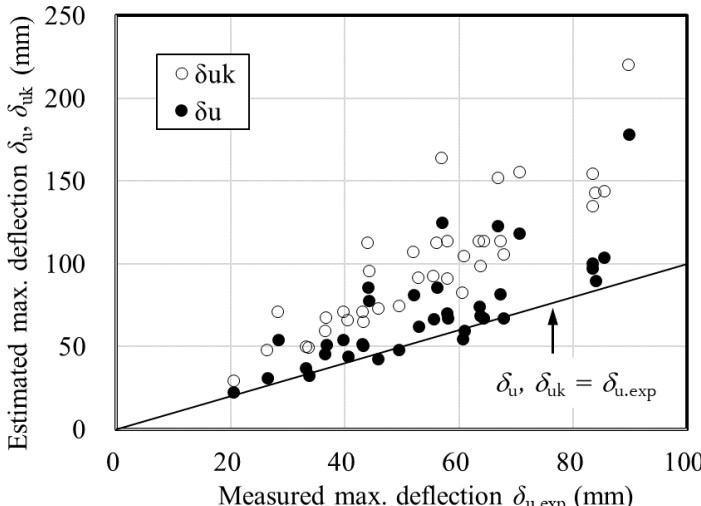

**Figure 6.** Comparison between the estimated maximum deflection $\delta_u$ and the measured value $\delta_{u.exp}$.

Consequently, $\delta_{u.exp}$ can be predicted considering the energy loss, due to the complete plastic collision between the weight and RC beam. This is because the target RC beam is a bending-failure type and is plastically deformed; therefore, the collision with the weight behaves like a nearly perfect plastic collision. On the other hand, in some cases, $\delta_u$ greatly exceeds $\delta_{u.exp}$. This is especially evident when the amount of deformation is large. It is assumed that this occurs because the RC beam has reached a region, in which the assumption of plane conservation does not hold. In the next section, based on this investigation, the scope of application of the plane-holding assumption is examined.

### 3.2. Scope of the Application of the RC Beam Plane Conservation Assumption

### 3.2.1. Examination Outline

To examine the applicable range of the plane conservation assumption of the RC beam, the experimental results of 134 cases of the bending-failure-type RC beam plastically deformed by a weight collision were collected. The specifications of the RC beams investigated in this study are shown in Appendix A (see Tables A1 and A2). The numerical values of the calculated bending capacity $P_u$ and shear capacity $Q_u$ were obtained from the literature, shown in those tables.

Table 2 shows the range of the considered specifications of the RC beams for the 134 cases. In this table, $\alpha$ is the shear-bending capacity ratio obtained by dividing the calculated shear capacity $Q_u$ by the calculated bending capacity $P_u$. RC beams with $\alpha \geq 1.0$ were selected here. In addition, when the velocity of the impactor is higher than approximately 80 m/s, the RC member is often damaged by penetration or perforation, including shear failure and backside spalling, prior to exhibiting bending deformation [14]. Moreover, it was reported that the maximum impact velocity of falling rocks is approximately 25 m/s [3]. Thus, those experiments with impact velocities of less than 25 m/s were considered. The diameter of the impactor is almost the same as the width of the RC beam, and the shape of the bottom surface of the weight was spherical with a small curvature. Each study confirmed that bending deformation was predominantly observed for all RC beams. Figure 7 shows the dimensions for the typical RC beams, considered here.

**Table 2.** Range of RC beam specifications (total of 134 cases).

| Item of Specification | Symbol | Unit | Range of Value | |
|---|---|---|---|---|
| | | | Min. | Max. |
| Size of section | | mm | $60 \times 100$ | $1000 \times 1,000$ |
| Clear span length | $L$ | m | 0.9 | 8.0 |
| Tensile rebar ratio | $p_t$ | % | 0.26 | 1.26 |
| Compressive strength of concrete | $f'_c$ | MPa | 25.2 | 52.0 |
| Yield strength of the tensile rebar | $f_y$ | Mpa | 235 | 520 |
| Calculated bending capacity | $P_u$ | kN | 4.10 | 881 |
| Calculated shear capacity | $Q_u$ | kN | 7.18 | 2,882 |
| Shear-bending capacity ratio | $\alpha$ | | 1.19 | 8.78 |
| Mass of beam | $M_b$ | t | 0.014 | 20.0 |
| Mass of weight | $M_w$ | t | 0.020 | 10.0 |
| Impact velocity of weight | $V$ | m/s | 1.00 | 19.8 |
| The kinetic energy of weight (input energy) | $E_k$ | kJ | 0.05 | 392.2 |

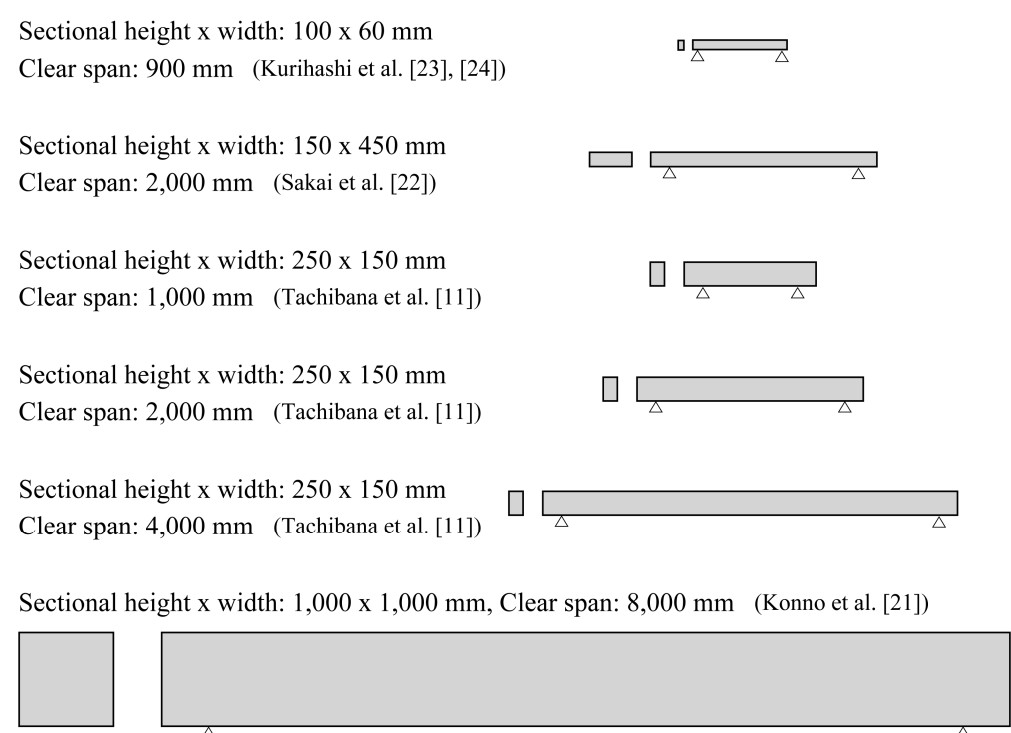

Sectional height x width: 100 x 60 mm
Clear span: 900 mm   (Kurihashi et al. [23], [24])

Sectional height x width: 150 x 450 mm
Clear span: 2,000 mm   (Sakai et al. [22])

Sectional height x width: 250 x 150 mm
Clear span: 1,000 mm   (Tachibana et al. [11])

Sectional height x width: 250 x 150 mm
Clear span: 2,000 mm   (Tachibana et al. [11])

Sectional height x width: 250 x 150 mm
Clear span: 4,000 mm   (Tachibana et al. [11])

Sectional height x width: 1,000 x 1,000 mm, Clear span: 8,000 mm   (Konno et al. [21])

**Figure 7.** Dimensions of typical RC beams that were considered in this study.

Figure 8 shows the relationship between the estimated maximum deflection $\delta_u$ and the experimental value $\delta_{u.exp}$. From the figure, the estimated value $\delta_u$ up to approximately 40 mm corresponds well with the experimental value $\delta_{u.exp}$. On the other hand, when $\delta_u$ becomes large, $\delta_u$ may overestimate the experimental ones. It is assumed to be because the RC beams have greatly deformed and damaged, so that the assumption of plane conservation does not hold as in the case of Figure 6.

However, as the minimum and maximum clear spans of the considered RC beam are 0.9 m and 8 m, respectively, the maximum deflection is expected to be significantly different even if the degree of damage is the same. Therefore, the accuracy of the estimated values and the applicable range must be examined without the results being affected by the shape and size of the beam.

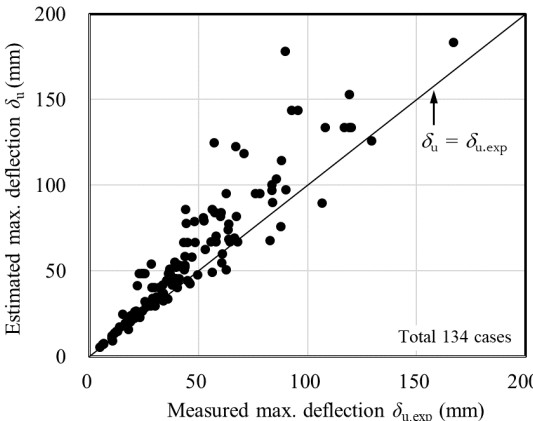

**Figure 8.** Comparison between the estimated and measured maximum deflections.

### 3.2.2. Investigation of the Estimation Accuracy Based on the Deflection Ratio $R_D$

In this section, the accuracy of the estimation method is assessed using the deflection ratio without considering the influence of the shape and dimensions of the beams. The value obtained by dividing the maximum deflection $\delta_u$ by the clear span length $L$ was defined as the deflection ratio $R_D$.

$$R_D = \delta_u/L \tag{18}$$

Figure 9 shows the relationship between the experimental and estimated deflection ratios. The figure shows that the estimated value of $R_D$ overestimates the experimental value $R_{D.exp}$ when 2% < $R_D$ < 9%. In contrast, when $R_D$ is larger than 9%, the estimated $R_D$ value corresponds relatively well to the experimental value $R_{D.exp}$. Therefore, the accuracy and the applicable range of the estimated values are difficult to examine based on the deflection ratio $R_D$.

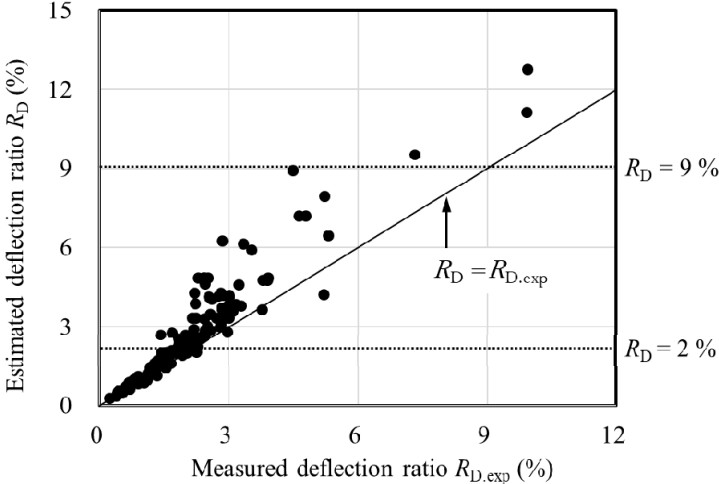

**Figure 9.** Investigation of the estimation accuracy based on the deflection ratio $R_D$.

### 3.2.3. Investigation of the Estimation Accuracy Based on the Plasticity Ratio $R_p$

The value obtained by dividing the maximum deflection $\delta_u$ by the tensile rebar yield deflection, $\delta_y$, is defined as the plasticity ratio $R_p$. It is an index generally used to evaluate ductility in the seismic design of RC piers.

$$R_p = \delta_u/\delta_y \tag{19}$$

Figure 10 shows the relationship between the estimated and experimental values of the plasticity ratio. The figure shows that when the plasticity ratio $R_p$ is large, the estimated ratio is higher than

the experimental result $R_{p.exp}$. It is observed that the maximum deflection $\delta_u$ can be estimated with relatively high accuracy when the plasticity ratio $R_p \leq 10$. In addition, if $R_p > 10$, the estimation accuracy is low. A high plasticity ratio indicates that the RC beam has a high degree of damage.

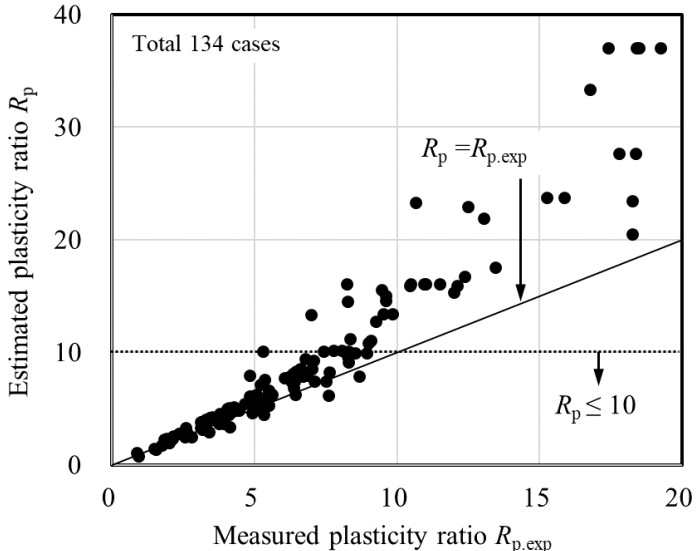

**Figure 10.** Investigation of the estimation accuracy based on the plasticity ratio $R_p$.

The estimated value of $R_p$ is calculated by assuming a bending deformation based on the plane conservation of the section of the RC beam in the calculation. Therefore, it is assumed that all transmitted energy is consumed by the bending deformation of the beam. In contrast, in the experiment, energy is usually consumed by not only the bending deformation, but also damage to the concrete at the collision point, pull-out of the reinforcing steel, and opening of shear cracks. Therefore, the estimated $R_p$ tended to be higher than $R_{p.exp}$ when the plasticity ratio $R_p$ was high.

*3.3. Investigation of the Accuracy and Applicable Range of the Deflection Estimation Formula*

The accuracy of the maximum deflection, $\delta_u$, can be calculated using Equation (20), and its application range is discussed below. The accuracy evaluation indicator (AEI) is defined as follows:

$$\text{AEI} = \delta_u/\delta_{u.exp}. \tag{20}$$

Figure 11 shows the relationship between the AEI and plasticity ratio for the design $R_p$ (= $\delta_u/\delta_y$) in the same manner as that in Reference [14]. From the figure, many plots can be obtained when the plasticity ratio $R_p \leq 10$, and the AEI is densely distributed between 0.8 and 1.6. Conversely, when the plasticity ratio $R_p > 10$, the AEI is widely distributed between 1.0 and 2.0.

Therefore, the application range of the proposed formula can be determined as $1 < R_p \leq 10$. When $R_p \leq 1$, the deflection of the RC beam is within the elastic range, and it does not exhibit a complete plastic collision. Therefore, $R_p < 1$ was excluded from the above application range.

To examine the estimation accuracy, Figure 12 shows the AEI when $1 < R_p \leq 10$. As shown in the figure, the AEI was distributed between 0.81 and 1.56, with an average value of 1.15, and a coefficient of variation of 0.113.

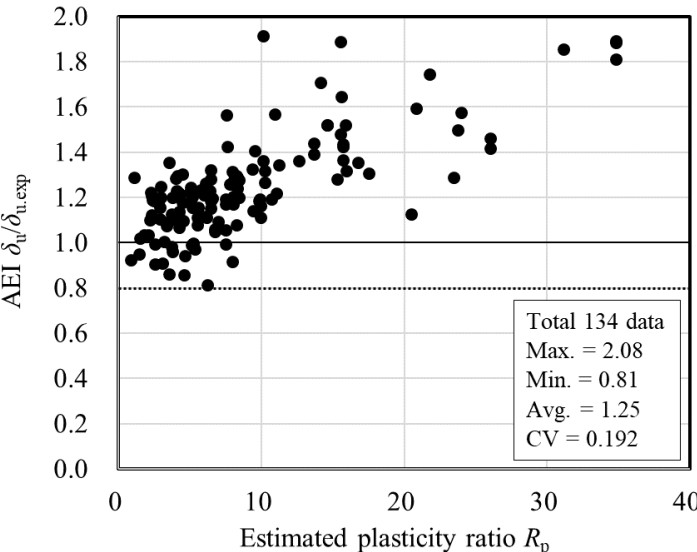

**Figure 11.** Accuracy of the estimated maximum deflection for the design $\delta_{ud}$ ($R_p < 40$).

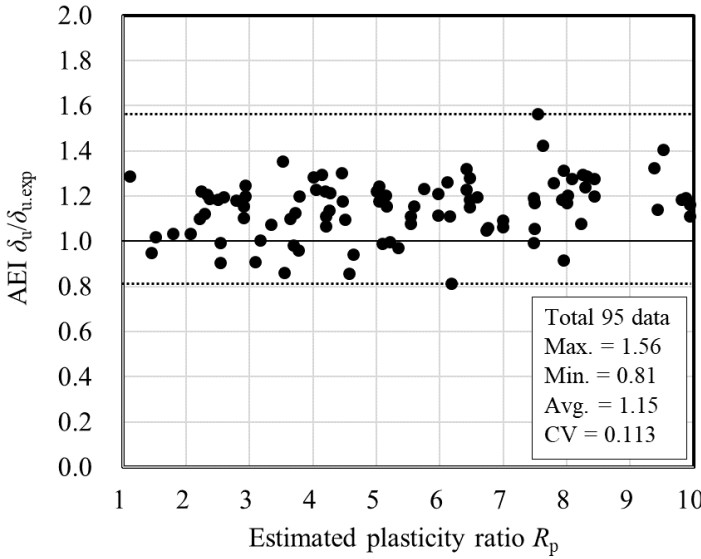

**Figure 12.** Accuracy of the estimated maximum deflection for the design $\delta_{ud}$ ($1 < R_p \leq 10$).

## 4. Simplified Estimation Method for the Maximum Deflection of the RC Beam Subjected to an Impact Load

Thus far, we have proposed a method for estimating the maximum deflection based on the assumption that the energy $E_t$ transmitted to the beam by a weight collision is equivalent to the absorption energy $E_a$, due to the bending deformation of the RC beam. Here, if the load–deflection curve of the RC beam can be simplified to a bilinear model, as shown in Figure 13, $E_a$ can be easily calculated, as expressed in Equation (21).

$$E_a = P_y\, \delta_{ud} - P_y\, \delta_y/2 \tag{21}$$

The deflection, when $E_a$ corresponds to $E_t$, is the maximum deflection for the design of $\delta_{ud}$. Therefore, $\delta_{ud}$ can be estimated as follows:

$$E_t = P_y\, \delta_{ud} - P_y\, \delta_y/2 \tag{22}$$

$$\delta_{ud} = E_t/P_y + \delta_y/2 \tag{23}$$

$$\delta_{ud} = \frac{M_w^2 V^2}{2P_y\left(\frac{17}{35}\rho AL + M_w\right)} + \frac{\delta_y}{2} \tag{24}$$

$$1 < R_p \leq 10 \tag{25}$$

To confirm the difference between $\delta_{ud}$ and $\delta_u$, Figure 14 illustrates the relationship between these values. The figure shows that $\delta_{ud}$ is nearly equivalent to $\delta_u$. Therefore, the maximum deflection $\delta_{ud}$ can be calculated using the transmitted energy $E_t$, yield load $P_y$ of the RC beam, and yield deflection $\delta_y$. Such an evaluation is possible when the relationship between the load and deflection is nearly bilinear, as in the case of a single bar RC beam. However, if the load–deflection relationship is composed of a curved line or multi-line, such as a prestressed concrete (PC) beam and other composite structural members, further investigations are required.

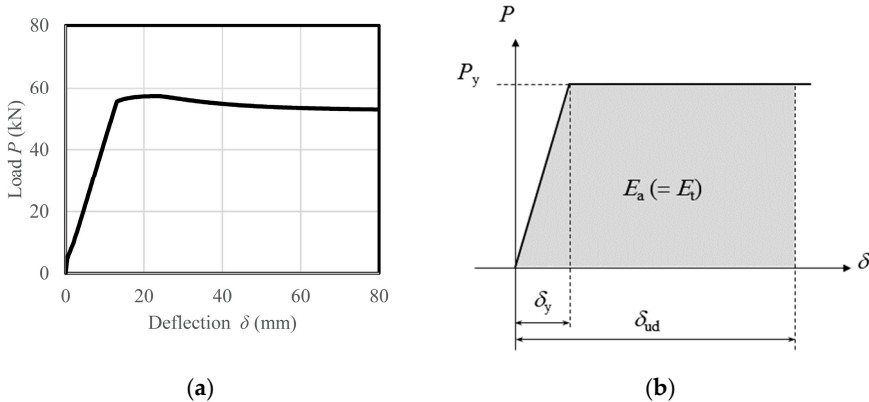

(a) (b)

**Figure 13.** Simplified load (*P*)–deflection (*δ*) relation of the RC beam; (**a**) Calculated *P*–*δ* relation (reprint of Figure 3), (**b**) Simplified *P*–*δ* relation.

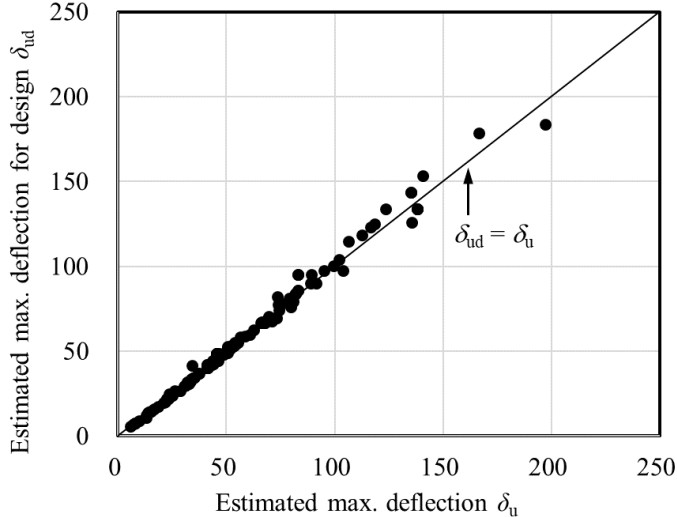

**Figure 14.** Relationship between $\delta_{ud}$ and $\delta_u$.

## 5. Conclusions

A method to estimate the maximum deflection of a bending-failure-type RC beam subjected to a collision action was established in this study. This method is based on the energy conservation concept where the transmitted energy acting on the beam by the weight collision is equivalent to the absorbed energy of the beam. The method was proposed based on reliable experimental results, and the validity and applicable range of this method were subsequently examined using the previous test results of 134 cases. The findings of this study are as follows:

(1)    The maximum deflection can be estimated with relatively high accuracy by using the transmitted impact energy obtained by assuming a vibration mode equal to the primary bending mode of the beam and a perfect plastic collision;

(2)    However, if the deflection is large, the estimated value overestimates the experimental value. This is thought to be because the assumption of the plane conservation of the cross-section of the RC beam does not hold;

(3)    Regardless of the shape and dimensions of the RC beam, if the estimated value of the plasticity ratio exceeds approximately 10, the assumption of plane conservation tends to fail;

(4)    A simplified estimation method for the maximum deflection was proposed by modeling the load–deflection relationship of the RC beams in a bilinear form. Assuming that the range of the plasticity ratio $R_p$ is from 1 to 10, the estimated value is approximately 15% larger than the experimental value. The coefficient of variation was approximately 0.11.

**Author Contributions:** Original draft preparation, Y.K.; review and editing, H.M. All authors have read and agreed to the published version of the manuscript.

**Funding:** This research received JSPS KAKENHI Grant Number 19H02394.

**Acknowledgments:** This study was compiled as part of the activities of the Impact Committee of the Structural Engineering Committee of Japan Society of Civil Engineers. While writing the paper, the committee members conducted literature surveys and numerous discussions.

**Conflicts of Interest:** The authors declare no conflict of interest.

## Symbols

| | |
|---|---|
| $f'_c$ | Compressive strength of concrete |
| $f_y$ | Yield strength of the tensile rebar |
| $p_t$ | Tensile rebar ratio |
| $E_k$ | Kinetic energy of weight (Input energy) |
| $E_t$ | Transmitted energy |
| L | Clear span of the beam |
| $M_b$ | Mass of beam |
| $M_{be}$ | Equivalent mass of beam |
| $M_w$ | Mass of weight |
| N | Amount of the rebar |
| $P_y$ | Calculated yield capacity of the beam |
| $P_u$ | Calculated bending capacity |
| $Q_u$ | Calculated shear capacity |
| $R_D$ | Deflection ratio ($=\delta_u/L$) |
| $R_{D.exp}$ | Experimental result of the deflection ratio ($=\delta_{u.exp}/L$) |
| $R_p$ | Plasticity ratio ($=\delta_u/\delta_y$) |
| $R_{p.exp}$ | Experimental result of the plasticity ratio ($=\delta_{u.exp}/\delta_y$) |
| V | Impact velocity of weight |
| $\alpha$ | Shear-bending capacity ratio ($=Q_u/P_u$) |
| $\delta_u$ | Estimated maximum deflection |
| $\delta_{ud}$ | Estimated maximum deflection for design |
| $\delta_{u.exp}$ | Experimental result of maximum deflection |
| $\delta_{uk}$ | Estimated maximum deflection based on the $E_k$ |
| $\delta_y$ | Calculated yield deflection of the beam |
| $\varphi$ | Nominal diameter of tensile rebar |

## Appendix A

The specifications of the RC beams investigated in this study are listed in Table A1. Maximum deflections $\delta_u$ estimated by the proposed method are listed in Table A2.

**Table A1.** Test specimens were used for examination.

| Name | Size of Section | Rebar φ (mm) | N | L (m) | $p_t$ (%) | $f'_c$ (MPa) | $f_y$ (MPa) | $P_u$ (kN) | $Q_u$ (kN) | $\alpha$ | $M_b$ (t) | $M_{be}$ (t) | $M_w$ (t) | V (m/s) | $E_k$ (kJ) | $E_t$ (kJ) |
|---|---|---|---|---|---|---|---|---|---|---|---|---|---|---|---|---|
| | | | | | | | | Tachibana et al. [11] | | | | | | | | |
| 1-A2 | | | | | | | | | | 0.15 | | | | 3.50 | 0.92 | 0.57 |
| 2-A2 | | | | | | | | | | 0.3 | | | | 2.40 | 0.86 | 0.66 |
| 3-A2 | | | | | | | | | | 0.45 | | | | 2.00 | 0.90 | 0.75 |
| 4-A2 | | | | | | | | | | 0.15 | | | | 4.90 | 1.80 | 1.12 |
| 5-A2 | | | | | | | | | | 0.3 | | | | 3.50 | 1.84 | 1.41 |
| 6-A2 | | | | | | | | | | 0.45 | | | | 2.80 | 1.76 | 1.47 |
| 7-A2 | | | | | | | | | | 0.15 | | | | 6.00 | 2.70 | 1.68 |
| 8-A2 | | | | | | | | | | 0.3 | | | | 4.20 | 2.65 | 2.03 |
| 9-A2 | | | | | | | | | | 0.45 | | | | 3.50 | 2.76 | 2.29 |
| 10-A2 | | | | 2.0 | | | | 33.3 | 91.1 | 2.74 | 0.188 | 0.091 | 0.3 | 1.00 | 0.15 | 0.12 |
| 11-A2-1 | | | | | | | | | | | | | | | | |
| 11-A2-2 | | | | | | | | | | | | | 0.3 | 2.00 | 0.60 | 0.46 |
| 12-A2-1 | 150 × 250 | 13 | 2 | | 0.8 | 31.0 | 383 | | | | | | | | | |
| 12-A2-2 | | | | | | | | | | | | | 0.3 | 3.00 | 1.35 | 1.04 |
| 13-A2-1 | | | | | | | | | | | | | | | | |
| 13-A2-2 | | | | | | | | | | | | | 0.3 | 4.00 | 2.40 | 1.84 |
| 14-A2-1 | | | | | | | | | | | | | | | | |
| 14-A2-2 | | | | | | | | | | | | | 0.3 | 5.00 | 3.75 | 2.88 |
| 14-A2-3 | | | | | | | | | | | | | | | | |
| 15-A1-1 | | | | | | | | | | | | | | | | |
| 15-A1-2 | | | | 1.0 | | | | 66.7 | 91.1 | 1.37 | 0.094 | 0.046 | 0.3 | 5.00 | 3.75 | 3.26 |
| 15-A1-3 | | | | | | | | | | | | | | | | |
| 15-A1-4 | | | | | | | | | | | | | | | | |
| 16-A4-1 | | | | | | | | | | | | | | | | |
| 16-A4-2 | | | | 4.0 | | | | 16.7 | 91.1 | 5.46 | 0.375 | 0.182 | 0.3 | 5.00 | 3.75 | 2.33 |
| 16-A4-3 | | | | | | | | | | | | | | | | |
| 17-B-1 17-B-2 | 300 × 150 | 13 | 4 | 2.0 | 1.53 | | 378 | 31.8 | 65.9 | 2.07 | 0.225 | 0.109 | 0.3 | 5.00 | 3.75 | 2.75 |
| 18-C-1 18-C-2 | 150 × 250 | 16 | 2 | 2.0 | 1.26 | | 402 | 50.3 | 94.8 | 1.88 | 0.188 | 0.091 | 0.3 | 5.00 | 3.75 | 2.88 |
| 19-D-1 19-D-2 | 150 × 250 | 10 | 2 | 2.0 | 0.45 | 27.3 | 393 | 20.2 | 87.1 | 4.31 | 0.188 | 0.091 | 0.3 | 5.00 | 3.75 | 2.88 |
| 20-E-1 20-E-2 | 150 × 400 | 13 | 2 | 2.0 | 0.47 | | 378 | 59.5 | 145.6 | 2.45 | 0.3 | 0.146 | 0.3 | 5.00 | 3.75 | 2.52 |
| 21-F-1 21-F-2 | 150 × 400 | 10 | 2 | 2.0 | 0.26 | | 393 | 34.9 | 140.6 | 4.03 | 0.3 | 0.146 | 0.3 | 5.00 | 3.75 | 2.52 |

**Table A1.** *Cont.*

| Name | Size of Section | Rebar φ (mm) | N | L (m) | $p_t$ (%) | $f'_c$ (MPa) | $f_y$ (MPa) | $P_u$ (kN) | $Q_u$ (kN) | α | $M_b$ (t) | $M_{be}$ (t) | $M_w$ (t) | V (m/s) | $E_k$ (kJ) | $E_t$ (kJ) |
|---|---|---|---|---|---|---|---|---|---|---|---|---|---|---|---|---|
| | | | | | | | Fujikake et al. [12] | | | | | | | | | |
| S1616-1 | | | | | | | | | | | | | | 1.72 | 0.59 | 0.51 |
| S1616-2 | | 16 | 2 | 1.4 | 1.26 | | 426 | 91.1 | 232 | 2.55 | 0.131 | 0.064 | 0.4 | 2.43 | 1.18 | 1.02 |
| S1616-3 | | | | | | | | | | | | | | 3.43 | 2.35 | 2.03 |
| S1616-4 | | | | | | | | | | | | | | 4.85 | 4.70 | 4.06 |
| S1322-1 | | | | | | | | | | | | | | 2.43 | 1.18 | 1.02 |
| S1322-2 | 150 × 250 | | | | | 42 | | | | | | | | 3.43 | 2.35 | 2.03 |
| S1322-3 | | | | | | | | | | | | | | 4.85 | 4.70 | 4.06 |
| S1322-4 | | 22 | 2 | 1.4 | 2.46 | | 418 | 162 | 245.4 | 1.51 | 0.131 | 0.064 | 0.4 | 6.86 | 9.41 | 8.12 |
| S2222-1 | | | | | | | | | | | | | | 2.43 | 1.18 | 1.02 |
| S2222-2 | | | | | | | | | | | | | | 3.43 | 2.35 | 2.03 |
| S2222-3 | | | | | | | | | | | | | | 4.85 | 4.70 | 4.06 |
| S2222-4 | | | | | | | | | | | | | | 6.86 | 9.41 | 8.12 |
| | | | | | | | Kishi et al. [13] | | | | | | | | | |
| PB-880 | | | | | | | | | | | | | | 9.90 | 98.07 | 16.74 |
| PB-880 | 1000 × 1000 | 32 | 7 | 8.0 | 0.65 | 33.3 | 382 | 881 | 2882 | 3.27 | 20.00 | 9.71 | 2.0 | 14.00 | 196.1 | 33.49 |
| PB-880 | | | | | | | | | | | | | | 17.15 | 294.2 | 50.23 |
| PB-880 | | | | | | | | | | | | | | 19.81 | 392.2 | 66.97 |
| PC-620 | 1000 × 850 | 29 | 7 | 8.0 | 0.64 | 31.2 | 400 | 621 | 1794 | 2.89 | 17.00 | 8.26 | 2.0 | 9.90 | 98.07 | 19.12 |
| PC-620 | | | | | | | | | | | | | | 14.00 | 196.1 | 38.24 |
| | | | | | | | Pham et al. [16] | | | | | | | | | |
| Beam 1 | 150 × 250 | 10 | 2 | 1.9 | 0.46 | 46 | 500 | 54.5 | 306.3 | 5.62 | 0.178 | 0.087 | 0.2035 | 6.26 | 3.99 | 2.80 |
| Beam 2 | | | | | | 52 | | 54.2 | 476.1 | 8.78 | | | | 6.26 | 3.99 | 2.80 |
| | | | | | | | Konno et al. [21] | | | | | | | | | |
| W2H10 | | | | | | | | | | | | | 2.0 | 14.00 | 196.1 | 33.49 |
| W5H4 | | | | | | | | | | | | | 5.0 | 8.86 | 196.1 | 66.65 |
| W10H2 | 1000 × 1000 | 25 | 7 | 8.0 | 0.42 | 29.2 | 382 | 613 | 2002 | 3.27 | 20.00 | 9.714 | 10.0 | 6.26 | 196.1 | 99.49 |
| W2H5 | | | | | | | | | | | | | 2.0 | 9.90 | 98.07 | 16.74 |
| W10H1 | | | | | | | | | | | | | 10.0 | 4.43 | 98.07 | 49.74 |

**Table A1.** *Cont.*

| Name | Size of Section | Rebar $\phi$ (mm) | N | $L$ (m) | $p_t$ (%) | $f'_c$ (MPa) | $f_y$ (MPa) | $P_u$ (kN) | $Q_u$ (kN) | $\alpha$ | $M_b$ (t) | $M_{be}$ (t) | $M_w$ (t) | $V$ (m/s) | $E_k$ (kJ) | $E_t$ (kJ) |
|---|---|---|---|---|---|---|---|---|---|---|---|---|---|---|---|---|
| Sakai et al. [22] | | | | | | | | | | | | | | | | |
| N-H0.1 | | | | | | | | | | | | | | 1.24 | 0.23 | 0.15 |
| N-H0.25 | | | | | | | | | | | | | | 2.06 | 0.64 | 0.41 |
| N-H0.5 | 450 × 150 | 13 | 4 | 2.0 | 1.02 | 25.2 | 383 | 39.9 | 87.8 | 2.20 | 0.338 | 0.164 | 0.3 | 2.98 | 1.33 | 0.86 |
| N-H1.0 | | | | | | | | | | | | | | 4.20 | 2.65 | 1.71 |
| N-H1.5 | | | | | | | | | | | | | | 5.13 | 3.95 | 2.55 |
| Kurihashi et al. [23,24] | | | | | | | | | | | | | | | | |
| N-H300 | | | | | | | | | | | | | | 2.30 | 0.05 | 0.04 |
| N-H600 | 60 × 100 | 6 | 1 | 0.9 | 0.75 | 31.6 | 380 | 4.1 | 7.18 | 1.75 | 0.014 | 0.007 | 0.02 | 3.20 | 0.10 | 0.08 |
| N-H900 | | | | | | | | | | | | | | 3.90 | 0.15 | 0.11 |
| Zhan et al. [25] | | | | | | | | | | | | | | | | |
| 6-C27-3 | | 6 | | 1.2 | 0.45 | | | 13.9 | 47.4 | 3.41 | 0.043 | 0.021 | 0.0336 | 7.67 | 0.99 | 0.61 |
| 6-C27-4 | | | | | | | | | | | | | | 8.85 | 1.32 | 0.81 |
| 8-C27-3 | | 8 | | 1.2 | 0.81 | | | 15.6 | 51.1 | 3.28 | 0.043 | 0.021 | 0.0336 | 7.67 | 0.99 | 0.61 |
| 8-C27-4 | | | | | | 27 | | | | | | | | 8.85 | 1.32 | 0.81 |
| 10-C27-2.5 | | | | | | | | | | | | | | 7.00 | 0.82 | 0.51 |
| 10-C27-3 | | 10 | | 1.2 | 1.26 | | | 27.2 | 54.4 | 2.00 | 0.043 | 0.021 | 0.0336 | 7.67 | 0.99 | 0.61 |
| 10-C27-4 | | | | | | | | | | | | | | 8.85 | 1.32 | 0.81 |
| 6-C40-5 | | | | | | | | | | | | | | 9.90 | 1.65 | 1.01 |
| 6-C40-6 | 120 × 120 | 6 | 6 | 1.2 | 0.45 | | 235 | 15.8 | 49.8 | 3.15 | 0.043 | 0.021 | 0.0336 | 10.84 | 1.97 | 1.22 |
| 6-C40-7 | | | | | | | | | | | | | | 11.71 | 2.30 | 1.42 |
| 6-C40-8 | | | | | | | | | | | | | | 12.52 | 2.63 | 1.62 |
| 8-C40-2 | | | | | | | | | | | | | | 6.26 | 0.66 | 0.41 |
| 8-C40-3 | | | | | | 40 | | | | | | | | 7.67 | 0.99 | 0.61 |
| 8-C40-4 | | 8 | | 1.2 | 0.81 | | | 19.4 | 54 | 2.79 | 0.043 | 0.021 | 0.0336 | 8.85 | 1.32 | 0.81 |
| 8-C40-5 | | | | | | | | | | | | | | 9.90 | 1.65 | 1.01 |
| 10-C40-5 | | | | | | | | | | | | | | 9.90 | 1.65 | 1.01 |
| 10-C40-6 | | 10 | | 1.2 | 1.26 | | | 29.9 | 57.8 | 1.93 | 0.043 | 0.021 | 0.0336 | 10.84 | 1.97 | 1.22 |
| 10-C40-7 | | | | | | | | | | | | | | 11.71 | 2.30 | 1.42 |

**Table A1.** *Cont.*

| Name | Size of Section | Rebar $\phi$ (mm) | N | L (m) | $p_t$ (%) | $f'_c$ (MPa) | $f_y$ (MPa) | $P_u$ (kN) | $Q_u$ (kN) | $\alpha$ | $M_b$ (t) | $M_{be}$ (t) | $M_w$ (t) | V (m/s) | $E_k$ (kJ) | $E_t$ (kJ) |
|---|---|---|---|---|---|---|---|---|---|---|---|---|---|---|---|---|
| Adhikary et al. [26] | | | | | | | | | | | | | | | | |
| DR3.8_0.8 _0.11_H0.6 | | | | | | | | 67.8 | 93.3 | 1.38 | 0.154 | 0.075 | | 3.43 | 1.76 | 1.41 |
| DR3.8_0.8 _0.11_H0.9 | 160 × 240 | 13 | 2 | 1.6 | 0.79 | 38.5 | 520 | | | | | | 0.3 | 4.20 | 2.65 | 2.12 |
| DR3.8_0.8 _0.11_H1.2 | | | | | | | | | | | | | | 4.85 | 3.53 | 2.83 |
| DR3.8_0.8 _0.15_H0.6 | | | | | | | | 67.8 | 102.3 | 1.51 | 0.154 | 0.075 | | 3.43 | 1.76 | 1.41 |
| DR3.8_0.8 _0.15_H0.9 | | | | | | | | | | | | | | 4.20 | 2.65 | 2.12 |
| DR3.8_0.8 _0.15_H1.2 | | | | | | | | | | | | | | 4.85 | 3.53 | 2.83 |
| DR5.7_1.6 _0.15_H0.3 | | | | | | | | 42.4 | 50.5 | 1.19 | 0.082 | 0.040 | | 2.43 | 0.89 | 0.78 |
| DR5.7_1.6 _0.15_H0.45 | 120 × 170 | 13 | 2 | 1.6 | 0.79 | 38.5 | 520 | | | | | | 0.3 | 2.97 | 1.32 | 1.17 |
| DR5.7_1.6 _0.15_H0.6 | | | | | | | | | | | | | | 3.43 | 1.76 | 1.56 |
| DR5.7_1.6 _0.20_H0.3 | | | | | | | | 42.4 | 56.4 | 1.33 | 0.082 | 0.040 | | 2.43 | 0.89 | 0.78 |
| DR5.7_1.6 _0.20_H0.45 | | | | | | | | | | | | | | 2.97 | 1.32 | 1.17 |
| DR5.7_1.6 _0.20_H0.6 | | | | | | | | | | | | | | 3.43 | 1.76 | 1.56 |

$\varphi$: Nominal diameter of tensile rebar, $N$: Amount of the rebar, $L$: Clear span length of the beam, $p_t$: Tensile rebar ratio, $f'_c$: Compressive strength of concrete, $f_y$: Yield strength of the tensile rebar, $P_u$: Calculated bending capacity, $Q_u$: Calculated shear capacity, $\alpha$: Shear-bending capacity ratio (= $Q_u/P_u$), $M_b$: Mass of beam, $M_{be}$: Equivalent mass of beam, $M_w$: Mass of weight, $V$: Impact velocity of weight, $E_k$: Kinetic energy of weight (Input energy), $E_t$: Transmitted energy.

**Table A2.** Estimated maximum deflection $\delta_u$.

| Ref. ID | Name | $\delta_{u.exp}$ (mm) | $E_k$ (kJ) | $E_t$ (kJ) | $\delta_{uk}$ (mm) | $\delta_u$ (mm) | $R_D$ (%) | $R_{D.exp}$ (%) | $R_p$ | $R_{p.exp}$ | $P_y$ (kN) | $\delta_y$ (mm) | $\delta_{ud}$ (mm) | AEI |
|---|---|---|---|---|---|---|---|---|---|---|---|---|---|---|
| | G1-1 | 64.3 | 7.35 | 4.25 | 114.0 | 67.3 | 2.24 | 2.14 | 7.42 | 7.09 | 67.1 | 9.07 | 67.9 | 1.06 |
| | G1-1S | 58.0 | 7.35 | 4.25 | 114.0 | 67.3 | 2.24 | 1.93 | 7.42 | 6.39 | 67.1 | 9.07 | 67.9 | 1.17 |
| | G2-1 | 28.3 | 2.40 | 1.84 | 71.2 | 54.0 | 2.70 | 1.42 | 10.09 | 5.29 | 35.8 | 5.35 | 54.1 | 1.91 |
| | G2-2 | 44.0 | 3.75 | 2.88 | 112.9 | 86.0 | 4.30 | 2.20 | 16.07 | 8.22 | 35.8 | 5.35 | 83.0 | 1.89 |
| | G2-3 | 57.0 | 5.40 | 4.14 | 164.1 | 124.9 | 6.25 | 2.85 | 23.35 | 10.65 | 35.8 | 5.35 | 118.4 | 2.08 |
| | G2L-1 | 44.2 | 3.20 | 2.61 | 95.9 | 77.7 | 3.89 | 2.21 | 14.52 | 8.26 | 35.8 | 5.35 | 75.5 | 1.71 |
| | G2L-2 | 66.8 | 5.00 | 4.07 | 151.7 | 122.9 | 6.15 | 3.34 | 22.97 | 12.49 | 35.8 | 5.35 | 116.4 | 1.74 |
| | G2L-3 | 89.7 | 7.20 | 5.86 | 220.2 | 178.5 | 8.93 | 4.49 | 33.36 | 16.77 | 35.8 | 5.35 | 166.5 | 1.86 |
| | G3-1 | 36.7 | 2.40 | 1.84 | 67.4 | 51.2 | 2.56 | 1.84 | 9.46 | 6.78 | 37.7 | 5.41 | 51.5 | 1.40 |
| | G3-2 | 52.0 | 3.75 | 2.88 | 107.0 | 81.4 | 4.07 | 2.60 | 15.05 | 9.61 | 37.7 | 5.41 | 79.0 | 1.52 |
| | G3-3 | 70.6 | 5.40 | 4.14 | 155.6 | 118.5 | 5.93 | 3.53 | 21.90 | 13.05 | 37.7 | 5.41 | 112.6 | 1.59 |
| | G4-1 | 39.7 | 2.40 | 1.84 | 71.2 | 54.0 | 2.70 | 1.99 | 10.09 | 7.42 | 35.8 | 5.35 | 54.1 | 1.36 |
| | G4-2 | 56.1 | 3.75 | 2.88 | 112.9 | 86.0 | 4.30 | 2.81 | 16.07 | 10.49 | 35.8 | 5.35 | 83.0 | 1.48 |
| | G5-1 | 63.5 | 7.20 | 4.66 | 113.9 | 74.2 | 2.47 | 2.12 | 7.99 | 6.84 | 66.8 | 9.29 | 74.3 | 1.17 |
| | G5-2 | 83.4 | 9.80 | 6.34 | 154.6 | 100.4 | 3.35 | 2.78 | 10.81 | 8.98 | 66.8 | 9.29 | 99.5 | 1.19 |
| | G6-1 | 26.4 | 3.75 | 2.49 | 47.7 | 30.9 | 1.55 | 1.32 | 5.46 | 4.66 | 83.8 | 5.66 | 32.5 | 1.23 |
| | G7-1 | 45.8 | 3.75 | 2.13 | 72.9 | 42.5 | 1.42 | 1.53 | 3.65 | 3.93 | 55.9 | 11.65 | 44.0 | 0.96 |
| 9 | G7-2 | 60.9 | 5.40 | 3.07 | 104.8 | 59.9 | 2.00 | 2.03 | 5.14 | 5.23 | 55.9 | 11.65 | 60.7 | 1.00 |
| | G8-1 | 36.5 | 5.40 | 4.08 | 59.4 | 45.5 | 2.28 | 1.83 | 5.10 | 4.09 | 101.5 | 8.93 | 44.7 | 1.22 |
| | G9-1 | 43.2 | 3.75 | 2.83 | 65.2 | 50.8 | 1.69 | 1.44 | 2.77 | 2.35 | 67.6 | 18.37 | 51.1 | 1.18 |
| | G9-2 | 57.9 | 5.40 | 4.08 | 91.4 | 70.5 | 2.35 | 1.93 | 3.84 | 3.15 | 67.6 | 18.37 | 69.5 | 1.20 |
| | G10-1 | 33.7 | 2.40 | 1.49 | 49.3 | 32.7 | 1.09 | 1.12 | 2.48 | 2.56 | 55.5 | 13.18 | 33.5 | 0.99 |
| | G10-2 | 49.5 | 3.75 | 2.33 | 74.6 | 48.0 | 1.60 | 1.65 | 3.64 | 3.76 | 55.5 | 13.18 | 48.6 | 0.98 |
| | G10-3 | 67.8 | 5.40 | 3.36 | 105.8 | 67.2 | 2.24 | 2.26 | 5.10 | 5.14 | 55.5 | 13.18 | 67.1 | 0.99 |
| | G10-4 | 83.9 | 7.35 | 4.57 | 142.9 | 90.1 | 3.00 | 2.80 | 6.84 | 6.37 | 55.5 | 13.18 | 89.0 | 1.06 |
| | G11-1 | 20.5 | 2.45 | 1.76 | 29.5 | 22.6 | 0.84 | 0.76 | 2.24 | 2.03 | 97.7 | 10.08 | 23.0 | 1.12 |
| | G11-2 | 33.2 | 4.41 | 3.16 | 50.1 | 36.9 | 1.37 | 1.23 | 3.66 | 3.29 | 97.7 | 10.08 | 37.4 | 1.13 |
| | G11-3 | 43.1 | 6.38 | 4.58 | 71.2 | 51.9 | 1.92 | 1.60 | 5.15 | 4.28 | 97.7 | 10.08 | 51.9 | 1.20 |
| | G11-4 | 55.5 | 8.35 | 5.99 | 92.5 | 67.0 | 2.48 | 2.06 | 6.65 | 5.51 | 97.7 | 10.08 | 66.4 | 1.20 |
| | G11-5 | 67.2 | 10.3 | 7.39 | 113.6 | 82.0 | 3.04 | 2.49 | 8.13 | 6.67 | 97.7 | 10.08 | 80.7 | 1.20 |
| | G11-6 | 83.4 | 12.2 | 8.79 | 134.7 | 97.3 | 3.60 | 3.09 | 9.65 | 8.27 | 97.7 | 10.08 | 95.0 | 1.14 |
| | G12-1 | 85.4 | 14.7 | 10.55 | 143.9 | 104.0 | 3.85 | 3.16 | 10.09 | 8.28 | 109.2 | 10.31 | 101.8 | 1.19 |
| | G13-1 | 60.6 | 14.7 | 9.65 | 82.4 | 54.9 | 2.03 | 2.24 | 7.87 | 8.68 | 185.6 | 6.98 | 55.5 | 0.92 |
| | G14-1 | 63.7 | 14.7 | 10.08 | 98.9 | 68.6 | 2.54 | 2.36 | 8.22 | 7.63 | 156.3 | 8.35 | 68.7 | 1.08 |
| | G15-1 | 40.5 | 14.7 | 9.65 | 65.9 | 44.3 | 1.64 | 1.50 | 6.06 | 5.54 | 233.7 | 7.31 | 44.9 | 1.11 |
| | G16-1 | 52.9 | 14.7 | 9.90 | 91.5 | 62.5 | 2.31 | 1.96 | 7.89 | 6.68 | 168.5 | 7.92 | 62.7 | 1.19 |

**Table A2.** *Cont.*

| Ref. ID | Name | $\delta_{u.exp}$ (mm) | $E_k$ (kJ) | $E_t$ (kJ) | $\delta_{uk}$ (mm) | $\delta_u$ (mm) | $R_D$ (%) | $R_{D.exp}$ (%) | $R_p$ | $R_{p.exp}$ | $P_y$ (kN) | $\delta_y$ (mm) | $\delta_{ud}$ (mm) | AEI |
|---|---|---|---|---|---|---|---|---|---|---|---|---|---|---|
| | 1-A2 | 13.6 | 0.92 | 0.57 | 26.6 | 17.4 | 0.87 | 0.68 | 3.33 | 2.60 | 36.2 | 5.23 | 18.4 | 1.35 |
| | 2-A2 | 16.3 | 0.86 | 0.66 | 25.0 | 19.8 | 0.99 | 0.82 | 3.79 | 3.12 | 36.2 | 5.23 | 20.9 | 1.28 |
| | 3-A2 | 17.9 | 0.90 | 0.75 | 26.0 | 22.0 | 1.10 | 0.90 | 4.21 | 3.42 | 36.2 | 5.23 | 23.3 | 1.30 |
| | 4-A2 | 25.4 | 1.80 | 1.12 | 51.9 | 32.1 | 1.61 | 1.27 | 6.14 | 4.86 | 36.2 | 5.23 | 33.6 | 1.32 |
| | 5-A2 | 31.6 | 1.84 | 1.41 | 53.1 | 40.5 | 2.03 | 1.58 | 7.74 | 6.04 | 36.2 | 5.23 | 41.6 | 1.31 |
| | 6-A2 | 33.3 | 1.76 | 1.47 | 50.7 | 42.1 | 2.11 | 1.67 | 8.05 | 6.37 | 36.2 | 5.23 | 43.1 | 1.30 |
| | 7-A2 | 37.0 | 2.70 | 1.68 | 78.8 | 48.4 | 2.42 | 1.85 | 9.25 | 7.07 | 36.2 | 5.23 | 49.0 | 1.32 |
| | 8-A2 | 43.7 | 2.65 | 2.03 | 77.3 | 58.7 | 2.94 | 2.19 | 11.22 | 8.36 | 36.2 | 5.23 | 58.7 | 1.34 |
| | 9-A2 | 48.4 | 2.76 | 2.29 | 80.7 | 66.6 | 3.33 | 2.42 | 12.73 | 9.25 | 36.2 | 5.23 | 65.9 | 1.36 |
| | 10-A2 | 4.50 | 0.15 | 0.12 | 6.60 | 5.70 | 0.29 | 0.23 | 1.09 | 0.86 | 36.2 | 5.23 | 5.79 | 1.29 |
| | 11-A2-1 | 12.8 | 0.60 | 0.46 | 18.2 | 14.7 | 0.74 | 0.64 | 2.81 | 2.45 | 36.2 | 5.23 | 15.3 | 1.20 |
| | 11-A2-2 | 12.3 | 0.60 | 0.46 | 18.2 | 14.7 | 0.74 | 0.62 | 2.81 | 2.35 | 36.2 | 5.23 | 15.3 | 1.25 |
| | 12-A2-1 | 25.8 | 1.35 | 1.04 | 38.7 | 29.8 | 1.49 | 1.29 | 5.70 | 4.93 | 36.2 | 5.23 | 31.2 | 1.21 |
| | 12-A2-2 | 28.0 | 1.35 | 1.04 | 38.7 | 29.8 | 1.49 | 1.40 | 5.70 | 5.35 | 36.2 | 5.23 | 31.2 | 1.12 |
| | 13-A2-1 | 42.2 | 2.40 | 1.84 | 69.8 | 53.1 | 2.66 | 2.11 | 10.15 | 8.07 | 36.2 | 5.23 | 53.5 | 1.27 |
| | 13-A2-2 | 40.6 | 2.40 | 1.84 | 69.8 | 53.1 | 2.66 | 2.03 | 10.15 | 7.76 | 36.2 | 5.23 | 53.5 | 1.32 |
| | 14-A2-1 | 60.1 | 3.75 | 2.88 | 110.5 | 84.2 | 4.21 | 3.01 | 16.10 | 11.49 | 36.2 | 5.23 | 82.1 | 1.37 |
| 11 | 14-A2-2 | 57.2 | 3.75 | 2.88 | 110.5 | 84.2 | 4.21 | 2.86 | 16.10 | 10.94 | 36.2 | 5.23 | 82.1 | 1.44 |
| | 14-A2-3 | 57.6 | 3.75 | 2.88 | 110.5 | 84.2 | 4.21 | 2.88 | 16.10 | 11.01 | 36.2 | 5.23 | 82.1 | 1.43 |
| | 15-A1-1 | 24.1 | 3.75 | 3.26 | 56.0 | 48.5 | 4.85 | 2.41 | 37.02 | 18.40 | 72.4 | 1.31 | 45.6 | 1.89 |
| | 15-A1-2 | 24.2 | 3.75 | 3.26 | 56.0 | 48.5 | 4.85 | 2.42 | 37.02 | 18.47 | 72.4 | 1.31 | 45.6 | 1.89 |
| | 15-A1-3 | 22.8 | 3.75 | 3.26 | 56.0 | 48.5 | 4.85 | 2.28 | 37.02 | 17.40 | 72.4 | 1.31 | 45.6 | 2.00 |
| | 15-A1-4 | 25.2 | 3.75 | 3.26 | 56.0 | 48.5 | 4.85 | 2.52 | 37.02 | 19.24 | 72.4 | 1.31 | 45.6 | 1.81 |
| | 16-A4-1 | 119.9 | 3.75 | 2.33 | 216.7 | 133.8 | 3.35 | 3.00 | 6.26 | 5.61 | 18.3 | 21.36 | 138.2 | 1.15 |
| | 16-A4-2 | 116.8 | 3.75 | 2.33 | 216.7 | 133.8 | 3.35 | 2.92 | 6.26 | 5.47 | 18.3 | 21.36 | 138.2 | 1.18 |
| | 16-A4-3 | 107.9 | 3.75 | 2.33 | 216.7 | 133.8 | 3.35 | 2.70 | 6.26 | 5.05 | 18.3 | 21.36 | 138.2 | 1.28 |
| | 17-B-1 | 78.0 | 3.75 | 2.75 | 132.2 | 95.3 | 4.77 | 3.90 | 8.04 | 6.58 | 35.7 | 11.85 | 82.9 | 1.06 |
| | 17-B-2 | 75.9 | 3.75 | 2.75 | 132.2 | 95.3 | 4.77 | 3.80 | 8.04 | 6.41 | 35.7 | 11.85 | 82.9 | 1.09 |
| | 18-C-1 | 41.1 | 3.75 | 2.88 | 69.2 | 53.3 | 2.67 | 2.06 | 8.57 | 6.61 | 58.3 | 6.22 | 52.5 | 1.28 |
| | 18-C-2 | 43.7 | 3.75 | 2.88 | 69.2 | 53.3 | 2.67 | 2.19 | 8.57 | 7.03 | 58.3 | 6.22 | 52.5 | 1.20 |
| | 19-D-1 | 95.5 | 3.75 | 2.88 | 189.3 | 143.7 | 7.19 | 4.78 | 27.63 | 18.37 | 21.7 | 5.20 | 135.2 | 1.42 |
| | 19-D-2 | 92.5 | 3.75 | 2.88 | 189.3 | 143.7 | 7.19 | 4.63 | 27.63 | 17.79 | 21.7 | 5.20 | 135.2 | 1.46 |
| | 20-E-1 | 29.6 | 3.75 | 2.52 | 60.1 | 40.3 | 2.02 | 1.48 | 13.39 | 9.83 | 63.7 | 3.01 | 41.1 | 1.39 |
| | 20-E-2 | 28.6 | 3.75 | 2.52 | 60.1 | 40.3 | 2.02 | 1.43 | 13.39 | 9.50 | 63.7 | 3.01 | 41.1 | 1.44 |
| | 21-F-1 | 43.0 | 3.75 | 2.52 | 100.7 | 66.9 | 3.35 | 2.15 | 23.72 | 15.25 | 38.1 | 2.82 | 67.7 | 1.57 |
| | 21-F-2 | 44.7 | 3.75 | 2.52 | 100.7 | 66.9 | 3.35 | 2.24 | 23.72 | 15.85 | 38.1 | 2.82 | 67.7 | 1.51 |

**Table A2.** *Cont.*

| Ref. ID | Name | $\delta_{u.exp}$ (mm) | $E_k$ (kJ) | $E_t$ (kJ) | $\delta_{uk}$ (mm) | $\delta_u$ (mm) | $R_D$ (%) | $R_{D.exp}$ (%) | $R_p$ | $R_{p.exp}$ | $P_y$ (kN) | $\delta_y$ (mm) | $\delta_{ud}$ (mm) | AEI |
|---|---|---|---|---|---|---|---|---|---|---|---|---|---|---|
| | S1616-1 | 5.80 | 0.59 | 0.51 | 7.80 | 6.90 | 0.49 | 0.41 | 2.37 | 1.99 | 89.2 | 2.91 | 7.18 | 1.24 |
| | S1616-2 | 10.2 | 1.18 | 1.02 | 14.2 | 12.4 | 0.89 | 0.73 | 4.26 | 3.51 | 89.2 | 2.91 | 12.9 | 1.26 |
| | S1616-3 | 18.8 | 2.35 | 2.03 | 28.0 | 24.2 | 1.73 | 1.34 | 8.32 | 6.46 | 89.2 | 2.91 | 24.2 | 1.29 |
| | S1616-4 | 36.0 | 4.70 | 4.06 | 56.5 | 48.7 | 3.48 | 2.57 | 16.74 | 12.37 | 89.2 | 2.91 | 46.9 | 1.30 |
| | S1322-1 | 6.30 | 1.18 | 1.02 | 8.70 | 7.70 | 0.55 | 0.45 | 2.24 | 1.83 | 165.7 | 3.44 | 7.87 | 1.25 |
| 12 | S1322-2 | 11.3 | 2.35 | 2.03 | 16.0 | 13.9 | 0.99 | 0.81 | 4.04 | 3.28 | 165.7 | 3.44 | 14.0 | 1.24 |
| | S1322-3 | 21.3 | 4.70 | 4.06 | 30.9 | 26.8 | 1.91 | 1.52 | 7.79 | 6.19 | 165.7 | 3.44 | 26.2 | 1.23 |
| | S1322-4 | 41.2 | 9.41 | 8.12 | 60.9 | 52.7 | 3.76 | 2.94 | 15.32 | 11.98 | 165.7 | 3.44 | 50.7 | 1.23 |
| | S2222-1 | 6.30 | 1.18 | 1.02 | 8.60 | 7.60 | 0.54 | 0.45 | 2.32 | 1.93 | 166.2 | 3.27 | 7.76 | 1.23 |
| | S2222-2 | 11.2 | 2.35 | 2.03 | 15.7 | 13.7 | 0.98 | 0.80 | 4.19 | 3.43 | 166.2 | 3.27 | 13.8 | 1.24 |
| | S2222-3 | 20.7 | 4.70 | 4.06 | 30.5 | 26.4 | 1.89 | 1.48 | 8.07 | 6.33 | 166.2 | 3.27 | 26.1 | 1.26 |
| | S2222-4 | 39.6 | 9.41 | 8.12 | 60.4 | 52.2 | 3.73 | 2.83 | 15.96 | 12.11 | 166.2 | 3.27 | 50.5 | 1.27 |
| | PB-880 | 29.8 | 98.07 | 16.74 | 123.2 | 29.5 | 0.37 | 0.37 | 1.48 | 1.49 | 823.6 | 20.00 | 30.3 | 1.02 |
| | PB-880 | 56.0 | 196.1 | 33.49 | 248.1 | 49.2 | 0.62 | 0.70 | 2.46 | 2.80 | 823.6 | 20.00 | 50.7 | 0.90 |
| 13 | PB-880 | 82.6 | 294.2 | 50.23 | 378.6 | 67.8 | 0.85 | 1.03 | 3.39 | 4.13 | 823.6 | 20.00 | 71.0 | 0.86 |
| | PB-880 | 106.6 | 392.3 | 66.97 | 510.1 | 89.9 | 1.12 | 1.33 | 4.50 | 5.33 | 823.6 | 20.00 | 91.3 | 0.86 |
| | PC-620 | 44.8 | 98.1 | 19.12 | 177.7 | 44.6 | 0.56 | 0.56 | 1.73 | 1.74 | 572.8 | 25.80 | 46.3 | 1.03 |
| | PC-620 | 87.7 | 196.1 | 38.24 | 370.7 | 76.1 | 0.95 | 1.10 | 2.95 | 3.40 | 572.8 | 25.80 | 79.7 | 0.91 |
| 16 | Beam 1 | 52.3 | 3.99 | 2.80 | 113.5 | 79.5 | 4.18 | 2.75 | 15.90 | 10.46 | 35.7 | 5.00 | 80.9 | 1.55 |
| | Beam 2 | 48.0 | 3.99 | 2.80 | 113.1 | 79.0 | 4.16 | 2.53 | 15.58 | 9.47 | 35.9 | 5.07 | 80.5 | 1.68 |
| | W2H10 | 66.4 | 196.1 | 33.49 | 377.5 | 69.5 | 0.87 | 0.83 | 3.45 | 3.30 | 531.2 | 20.12 | 73.1 | 1.10 |
| | W5H4 | 129.3 | 196.1 | 66.65 | 377.5 | 125.9 | 1.57 | 1.62 | 6.26 | 6.43 | 531.2 | 20.12 | 135.5 | 1.05 |
| 21 | W10H2 | 166.7 | 196.1 | 99.49 | 377.5 | 183.5 | 2.29 | 2.08 | 9.12 | 8.29 | 531.2 | 20.12 | 197.3 | 1.18 |
| | W2H5 | 40.2 | 98.0 | 16.74 | 180.7 | 40.3 | 0.50 | 0.50 | 2.00 | 2.00 | 531.2 | 20.12 | 41.6 | 1.03 |
| | W10H1 | 89.8 | 98.0 | 49.74 | 180.7 | 97.4 | 1.22 | 1.12 | 4.84 | 4.46 | 531.2 | 20.12 | 103.7 | 1.15 |

**Table A2.** *Cont.*

| Ref. ID | Name | $\delta_{u.exp}$ (mm) | $E_k$ (kJ) | $E_t$ (kJ) | $\delta_{uk}$ (mm) | $\delta_u$ (mm) | $R_D$ (%) | $R_{D.exp}$ (%) | $R_p$ | $R_{p.exp}$ | $P_y$ (kN) | $\delta_y$ (mm) | $\delta_{ud}$ (mm) | AEI |
|---|---|---|---|---|---|---|---|---|---|---|---|---|---|---|
| | N-H0.1 | 10.5 | 0.23 | 0.15 | 11.3 | 9.15 | 0.46 | 0.53 | 0.80 | 0.91 | 37.6 | 11.48 | 9.71 | 0.92 |
| | N-H0.25 | 17.6 | 0.64 | 0.41 | 21.6 | 16.0 | 0.80 | 0.88 | 1.39 | 1.53 | 37.6 | 11.48 | 16.7 | 0.95 |
| 22 | N-H0.5 | 24.2 | 1.33 | 0.86 | 39.9 | 26.9 | 1.35 | 1.21 | 2.34 | 2.11 | 37.6 | 11.48 | 28.6 | 1.18 |
| | N-H1.0 | 43.5 | 2.65 | 1.71 | 85.7 | 52.4 | 2.62 | 2.18 | 4.56 | 3.79 | 37.6 | 11.48 | 51.2 | 1.18 |
| | N-H1.5 | 59.9 | 3.95 | 2.55 | 133.0 | 82.1 | 4.11 | 3.00 | 7.15 | 5.22 | 37.6 | 11.48 | 73.6 | 1.23 |
| | N-H300 | 10.0 | 0.05 | 0.04 | 13.8 | 10.8 | 1.20 | 1.11 | 3.45 | 3.19 | 3.50 | 3.13 | 12.9 | 1.29 |
| 23, 24 | N-H600 | 15.1 | 0.10 | 0.08 | 35.1 | 24.9 | 2.77 | 1.68 | 7.96 | 4.82 | 3.50 | 3.13 | 23.6 | 1.56 |
| | N-H900 | 21.9 | 0.15 | 0.11 | 58.1 | 41.7 | 4.63 | 2.43 | 13.33 | 7.00 | 3.50 | 3.13 | 34.3 | 1.57 |
| | 6-C27-3 | 47.0 | 0.99 | 0.61 | 94.8 | 58.2 | 4.85 | 3.91 | 8.29 | 6.69 | 11.5 | 7.02 | 56.4 | 1.20 |
| | 6-C27-4 | 63.7 | 1.32 | 0.81 | 126.3 | 77.5 | 6.46 | 5.31 | 11.04 | 9.07 | 11.5 | 7.02 | 73.9 | 1.16 |
| | 8-C27-3 | 30.6 | 0.99 | 0.61 | 55.1 | 34.7 | 2.89 | 2.55 | 4.26 | 3.75 | 19.6 | 8.15 | 35.1 | 1.15 |
| | 8-C27-4 | 39.4 | 1.32 | 0.81 | 72.9 | 45.5 | 3.79 | 3.29 | 5.58 | 4.84 | 19.6 | 8.15 | 45.4 | 1.15 |
| | 10-C27-2.5 | 18.5 | 0.82 | 0.51 | 31.2 | 20.4 | 1.70 | 1.54 | 2.21 | 2.00 | 29.7 | 9.25 | 21.7 | 1.17 |
| | 10-C27-3 | 20.0 | 0.99 | 0.61 | 37.0 | 23.9 | 1.99 | 1.67 | 2.58 | 2.16 | 29.7 | 9.25 | 25.1 | 1.26 |
| | 10-C27-4 | 28.7 | 1.32 | 0.81 | 48.3 | 30.8 | 2.57 | 2.39 | 3.33 | 3.10 | 29.7 | 9.25 | 31.9 | 1.11 |
| | 6-C40-5 | 62.7 | 1.65 | 1.01 | 156.1 | 95.3 | 7.94 | 5.22 | 14.59 | 9.59 | 11.8 | 6.53 | 89.2 | 1.42 |
| 25 | 6-C40-6 | 87.7 | 1.97 | 1.22 | 186.7 | 114.5 | 9.54 | 7.31 | 17.53 | 13.44 | 11.8 | 6.53 | 106.2 | 1.21 |
| | 6-C40-7 | 119.1 | 2.30 | 1.42 | 218.2 | 133.9 | 11.16 | 9.92 | 20.51 | 18.23 | 11.8 | 6.53 | 123.4 | 1.04 |
| | 6-C40-8 | 119.1 | 2.63 | 1.62 | 249.8 | 153.3 | 12.78 | 9.93 | 23.48 | 18.24 | 11.8 | 6.53 | 140.6 | 1.18 |
| | 8-C40-2 | 22.9 | 0.66 | 0.41 | 36.4 | 23.0 | 1.92 | 1.91 | 3.17 | 3.16 | 20.2 | 7.26 | 23.7 | 1.03 |
| | 8-C40-3 | 35.7 | 0.99 | 0.61 | 54.0 | 33.6 | 2.80 | 2.97 | 4.63 | 4.91 | 20.2 | 7.26 | 33.7 | 0.95 |
| | 8-C40-4 | 35.1 | 1.32 | 0.81 | 71.8 | 44.4 | 3.70 | 2.93 | 6.12 | 4.84 | 20.2 | 7.26 | 43.7 | 1.24 |
| | 8-C40-5 | 38.8 | 1.65 | 1.01 | 89.5 | 55.3 | 4.61 | 3.24 | 7.62 | 5.35 | 20.2 | 7.26 | 53.8 | 1.39 |
| | 10-C40-5 | 33.7 | 1.65 | 1.01 | 58.8 | 37.0 | 3.08 | 2.81 | 4.50 | 4.09 | 30.6 | 8.23 | 37.2 | 1.11 |
| | 10-C40-6 | 45.2 | 1.97 | 1.22 | 69.9 | 43.9 | 3.66 | 3.77 | 5.33 | 5.49 | 30.6 | 8.23 | 43.8 | 0.97 |
| | 10-C40-7 | 62.5 | 2.30 | 1.42 | 82.3 | 50.9 | 4.24 | 5.21 | 6.18 | 7.59 | 30.6 | 8.23 | 50.5 | 0.81 |

**Table A2.** *Cont.*

| Ref. ID | Name | $\delta_{u.exp}$ (mm) | $E_k$ (kJ) | $E_t$ (kJ) | $\delta_{uk}$ (mm) | $\delta_u$ (mm) | $R_D$ (%) | $R_{D.exp}$ (%) | $R_p$ | $R_{p.exp}$ | $P_y$ (kN) | $\delta_y$ (mm) | $\delta_{ud}$ (mm) | AEI |
|---|---|---|---|---|---|---|---|---|---|---|---|---|---|---|
| 26 | DR3.8_0.8_0.11_H0.6 | 18.6 | 1.76 | 1.41 | 28.6 | 23.1 | 1.44 | 1.16 | 5.03 | 4.05 | 65.9 | 4.59 | 23.7 | 1.28 |
| | DR3.8_0.8_0.11_H0.9 | 34.5 | 2.65 | 2.12 | 42.7 | 34.3 | 2.14 | 2.16 | 7.47 | 7.52 | 65.9 | 4.59 | 34.5 | 1.00 |
| | DR3.8_0.8_0.11_H1.2 | 41.0 | 3.53 | 2.83 | 56.8 | 45.6 | 2.85 | 2.56 | 9.93 | 8.93 | 65.9 | 4.59 | 45.2 | 1.10 |
| | DR3.8_0.8_0.15_H0.6 | 19.6 | 1.76 | 1.41 | 28.6 | 23.1 | 1.44 | 1.23 | 5.03 | 4.27 | 65.9 | 4.59 | 23.7 | 1.21 |
| | DR3.8_0.8_0.15_H0.9 | 28.8 | 2.65 | 2.12 | 42.7 | 34.3 | 2.14 | 1.80 | 7.47 | 6.27 | 65.9 | 4.59 | 34.5 | 1.20 |
| | DR3.8_0.8_0.15_H1.2 | 39.2 | 3.53 | 2.83 | 56.8 | 45.6 | 2.85 | 2.45 | 9.93 | 8.54 | 65.9 | 4.59 | 45.2 | 1.15 |
| | DR5.7_1.6_0.15_H0.3 | 20.0 | 0.89 | 0.78 | 24.9 | 22.1 | 1.38 | 1.25 | 2.90 | 2.63 | 41.7 | 7.61 | 22.6 | 1.13 |
| | DR5.7_1.6_0.15_H0.45 | 30.0 | 1.32 | 1.17 | 35.9 | 32.0 | 2.00 | 1.88 | 4.20 | 3.94 | 41.7 | 7.61 | 31.8 | 1.06 |
| | DR5.7_1.6_0.15_H0.6 | 39.1 | 1.76 | 1.56 | 47.3 | 42.1 | 2.63 | 2.44 | 5.53 | 5.14 | 41.7 | 7.61 | 41.2 | 1.05 |
| | DR5.7_1.6_0.20_H0.3 | 19.1 | 0.89 | 0.78 | 24.9 | 22.1 | 1.38 | 1.19 | 2.90 | 2.51 | 41.7 | 7.61 | 22.6 | 1.18 |
| | DR5.7_1.6_0.20_H0.45 | 28.8 | 1.32 | 1.17 | 35.9 | 32.0 | 2.00 | 1.80 | 4.20 | 3.78 | 41.7 | 7.61 | 31.8 | 1.11 |
| | DR5.7_1.6_0.20_H0.6 | 37.9 | 1.76 | 1.56 | 47.3 | 42.1 | 2.63 | 2.37 | 5.53 | 4.98 | 41.7 | 7.61 | 41.2 | 1.09 |

$\delta_{u.exp}$: Experimental result of maximum deflection, $E_k$: Kinetic energy of weight (Input energy), $E_t$: Transmitted energy, $\delta_{uk}$: Estimated maximum deflection based on the $E_k$, $\delta_u$: Estimated maximum deflection, $R_D$: Deflection ratio (= $\delta_u$ /L), $R_{D.exp}$: Experimental result of the deflection ratio (= $\delta_{u.exp}$/L), $R_p$: Plasticity ratio (= $\delta_u/\delta_y$), $R_{p.exp}$: Experimental result of the plasticity ratio (= $\delta_{u.exp}$ /$\delta_y$), $P_y$: Calculated yield bending capacity of the beam, $\delta_y$: Calculated yield deflection of the beam, $\delta_{ud}$: Estimated maximum deflection for design, AEI: Accuracy evaluation indicator (= $\delta_u$ /$\delta_{u.exp}$).

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
