# Peer review of "Simplified Estimation Method for Maximum Deflection in Bending-Failure-Type Reinforced Concrete Beams Subjected to Collision Action and Its Application Range"

_applsci, doi:10.3390/app10196941_

Round 1

Reviewer 1 Report

Both Abstract  and  Conclusion part should be modifying, and strengthening to  data information & RC design. 

Reviewer 2 Report

In this paper, an estimation model for the deflection of RC beams
subjected to impact loading is proposed based on the energy conservation law.

Besides, a simple method for estimating the maximum deflection is proposed, in addition to a structural performance design method for RC  beams subjected to collision.

Interesting and actual topic. However  the employed mechanical models are unrealistic because  don't rely on well established scientific theories.The proposed method cannot be based solely on highly reliable experimental results, 

Based on plausible models, some researchers have recently proposed  simple formulas for the same problem.

In conclusion, it is necessary to revise the theoretical assumptions related to the proposed method.

Reviewer 3 Report

The manuscript seems to have a very good contribution. In the reviewer's opinion, it can be interesting for readerships. The manuscript is well organized, however, there are several Grammatical mistakes that should be resolved. Moreover, there are some technical and conceptual issues in the manuscript, which should be revised. The following comments are made to improve both content and structure. Accordingly, the manuscript should be revised through the application of the comments. 

1. The introduction of the manuscript should be definitely improved. The performance-based approach should be elaborated by mentioning more applied examples. This is indeed necessary to show the introduction of other simplified methods that have used the performance-based design approach. Authors are asked to review and mention the following articles in the introduction as additional references:

  • A. Rezaei Rad, M. Banazadeh, Probabilistic risk-based performance evaluation of seismically base-isolated steel structures subjected to far-field earthquakes, Buildings. 8 (2018). doi:10.3390/buildings8090128
  • Grubišić, J. Ivošević, A. Grubišić, Reliability Analysis of Reinforced Concrete Frame by Finite Element Method with Implicit Limit State Functions, Buildings. 9 (2019) 119. https://doi.org/10.3390/buildings9050119.

2. The authors should be aware of using appropriate articles (i.e. “the”, “an”, and “a”). Please review the text again and be sure you have used the appropriate structure. 

3. Overall, you should keep the same verb tense for each section. The literature review should be written in the past tense.

4. It is recommended to stick to the same voice in the text. It should be either passive or active. Double-check this in the manuscript.

5. Paragraphs show provide readerships with an independent statement. As such, they do not start with "However", "Nevertheless", "Since", etc. You should avoid making this error in your manuscript. Please see the beginning of line 74 as an example.

6. Line 135-136: " The numerical values for the calculated bending capacity Pu and the shear capacity Qu were obtained from the literature". Please refer to the corresponding literature, table, or figure.

7. Figure 8 is very important and the authors should pay much more attention to this section. Please provide more explanation for the observations you mentioned in lines 204-212. Specifically, "the applicable range of this estimation method must be examined" in line 210: please give a concrete range.    

8. The previous comment is equally applied to Figure 9-10.

Round 2

Reviewer 2 Report

The paper has been sufficiently revised. You need to better explain equations 4-6 (, =)

Reviewer 3 Report

I believe the edits are fine.
